# Src activates retrograde membrane traffic through phosphorylation of GBF1

**Joanne Chia[1]\*, Shyi-Chyi Wang[1,2], Sheena Wee[1], David James Gill[1], Felicia Tay[1], Srinivasaraghavan Kannan[3], Chandra S Verma[3,4,5], Jayantha Gunaratne[1], Frederic A Bard[1]\*†**

[1]Institute of Molecular and Cell Biology, Singapore, Singapore; [2]Institute of Bioengineering and Bioimaging, Singapore, Singapore; [3]Bioinformatics Institute, Singapore, Singapore; [4]Department of Biological Sciences, National University of Singapore, Singapore, Singapore; [5]School of Biological Sciences, Nanyang Technological University, Singapore, Singapore

**Abstract** The Src tyrosine kinase controls cancer-critical protein glycosylation through Golgi to ER relocation of GALNTs enzymes. How Src induces this trafficking event is unknown. Golgi to ER transport depends on the GTP exchange factor (GEF) GBF1 and small GTPase Arf1. Here, we show that Src induces the formation of tubular transport carriers containing GALNTs. The kinase phosphorylates GBF1 on 10 tyrosine residues; two of them, Y876 and Y898, are located near the C-terminus of the Sec7 GEF domain. Their phosphorylation promotes GBF1 binding to the GTPase; molecular modeling suggests partial melting of the Sec7 domain and intramolecular rearrangement. GBF1 mutants defective for these rearrangements prevent binding, carrier formation, and GALNTs relocation, while phosphomimetic GBF1 mutants induce tubules. In sum, Src promotes GALNTs relocation by promoting GBF1 binding to Arf1. Based on residue conservation, similar regulation of GEF-Arf complexes by tyrosine phosphorylation could be a conserved and widespread mechanism.

**\*For correspondence:**
zhchia@imcb.a-star.edu.sg (JC);
fbard@imcb.a-star.edu.sg (FAB)

**Present address:** †Centre de Recherche en Cancérologie de Marseille, CRCM, Equipe ARC Leader in Oncology and AMIDEX Chaire d'excellence, Aix Marseille Université, Inserm, CNRS, Institut Paoli-Calmettes, 13009, Marseille, France. frederic.bard@inserm.fr, Marseille, France

## Editor's evaluation

The Src tyrosine kinase controls cancer-critical protein glycosylation through Golgi to ER relocation of a subset of Golgi enzymes, GALNTs, from the Golgi to the ER. The authors show here that Src induces the formation of tubular transport carriers containing GALNTs by phosphorylating GBF1 and promoting its binding to Arf1. This study presents some of the first clues to the molecular events underlying Src-regulated relocalization of glycosyltransferases.

## Introduction

Eukaryotic cells constantly regulate membrane trafficking between compartments to adjust their physiology. Thus, signaling pathways impinge on trafficking pathways at many different levels. For instance, the Src tyrosine kinase has been shown to regulate Golgi membranes, in part to adjust trafficking rates in response to change in cargo load (*Pulvirenti et al., 2008*). Changes in Src activity have major effects on the morphology of the Golgi apparatus, with compaction after Src depletion or fragmentation upon Src hyper-activation (*Bard et al., 2003*; *Weller et al., 2010*).

A specific role of the Src kinase at the Golgi is to regulate protein O-glycosylation. Src activation induces the relocation of polypeptide GalNAc transferases (GALNTs) from the Golgi to the endoplasmic reticulum (ER) (*Bard and Chia, 2016*). GALNTs initiate GalNAc type O-glycosylation and their relocation leads to a marked increase in glycosylation, which can be measured with the levels of the Tn glycan (a single GalNac residue). Increase in Tn can be detected by lectins such as HPL and *Vicia*

*villosa* lectin (VVL) (*Gill et al., 2011*; *Gill et al., 2010*). The Tn levels' increase corresponds to multiple cell surface and ER resident proteins becoming hyper-O-glycosylated; such as MMP14, PDIA4, and Calnexin (*Nguyen et al., 2017*; *Ros et al., 2020*). In short, GALNTs relocation upregulates GalNac type O-glycosylation of many proteins.

This *GALNTs Activation* (GALA) pathway is strongly activated in breast, lung, and liver cancers and presumably in most high Tn-expressing tumors. GALA markedly promotes tumor growth and metastasis (*Gill et al., 2013*; *Nguyen et al., 2017*; *Ros et al., 2020*). In addition to Src, the pathway can be stimulated by the cell surface receptors EGFR or PDGFR and is controlled by a complex signaling network, including a constitutive negative regulation by the kinase ERK8 in some cell types (*Chia et al., 2014*). Src has long been implicated in tumorigenesis and tumor invasiveness, and GALA is likely an important mediator of Src oncogenic effects (*Chia et al., 2019*). It is unclear how Src stimulates the transport of GALNTs from the Golgi to the ER apart from the fact that the process involves the Arf1 small GTPase and can be blocked by a dominant negative form of Arf1 (*Gill et al., 2010*).

Arf1 is part of a family of small GTPases involved in many aspects of intracellular membranes (*D'Souza-Schorey and Chavrier, 2006*). Arfs function in conjunction with larger proteins called GTP exchange factors (GEFs) that mediate the transfer from GDP to GTP-bound form of the small GTPase. GEFs bind to the GDP-bound form of Arfs and not the GTP-bound form. All GEFs have in common a Sec7 domain (Sec7d) that specifically mediates displacement of GDP from Arf-GDP and loading with GTP. Thus, Arfs function like molecular timers, oscillating between GDP- and GTP-bound forms and binding different partners (*Cherfils, 2014*).

There are seven subfamilies of Arf GEFs in eukaryotes, with two subfamilies operating at the Golgi: BIG1/2 and GBF1 (*Cox et al., 2004*). While the BIGs primarily function at the *trans*-Golgi network and endosomal compartments, GBF1 functions at the early *cis*-Golgi and ER-Golgi intermediate compartment (ERGIC) and regulates Golgi to ER retrograde traffic (*Kawamoto et al., 2002*; *Zhao et al., 2006*; *Zhao et al., 2002*).

GBF1 functions mostly with Arf1. GBF1 contains five other conserved domains, two in N-terminal of the Sec7d (DCB and HUS) and three in C-terminal (HDS1 to 3). These domains are thought to mediate GBF1 recruitment to membranes and/or regulate membrane transport (*Richardson et al., 2012*). Work by Melançon's group has identified membrane-bound Arf-GDP as a factor regulating GBF1 recruitment to *cis*-Golgi membranes (*Quilty et al., 2018*; *Quilty et al., 2014*). In addition, they propose that the C-terminal domains HDS1 and 2 are required to bind to an unidentified Golgi receptor (*Quilty et al., 2018*). More recently, HDS1 has been shown to bind to phosphoinositides such as PIP3, PI4P, and PI(4,5)P2 (*Meissner et al., 2018*).

Here, we report the setup of an inducible Src activation system to rapidly and reliably activate the relocation of GALNTs to the ER. Acute activation of Src stimulates the formation of tubular-shaped, GALNTs-containing transport carriers and results in increased O-glycosylation levels. Src activation induces the recruitment of GBF1 at the Golgi, increased binding of GBF1 to Arf1, and a transient upregulation of Arf1-GTP levels. By mass spectrometry, we found that Src directly phosphorylates GBF1 on 10 residues, including residues Y876 and Y898, located within and close to the C-terminus of the Sec7d, respectively. In silico modeling and directed mutagenesis suggest important conformational changes that promote binding to Arf1. As supported by additional mutants, Y876 phosphorylation induces partial melting of an alpha-helix in Sec7d, increasing binding affinity to Arf1. Y898 phosphorylation appears to release an interaction between Sec7d and the linker domain in C-terminal. We propose that tyrosine phosphorylation increases GBF1 affinity for Arf-GDP on Golgi membranes and promotes the relocation of GALNTs through tubular transport intermediates.

## Results

### Src8A7F chemical activation induces rapid GALNTs relocation to the ER

ER relocation of GALNTs correlates well in solid tumors with total Tn levels, measured using staining with *Helix pomatia* lectin (HPL) (*Gill et al., 2010*; *Hammarström et al., 1977*; *Figure 1A*). High levels of relocation or GALA are found in a majority of samples from malignant tumors. By contrast and for unknown reasons, most cancer cell lines in vitro show limited levels of GALA (*Gill et al., 2013*). A more marked relocation can be induced in cell lines by transfecting a plasmid expressing an active form of

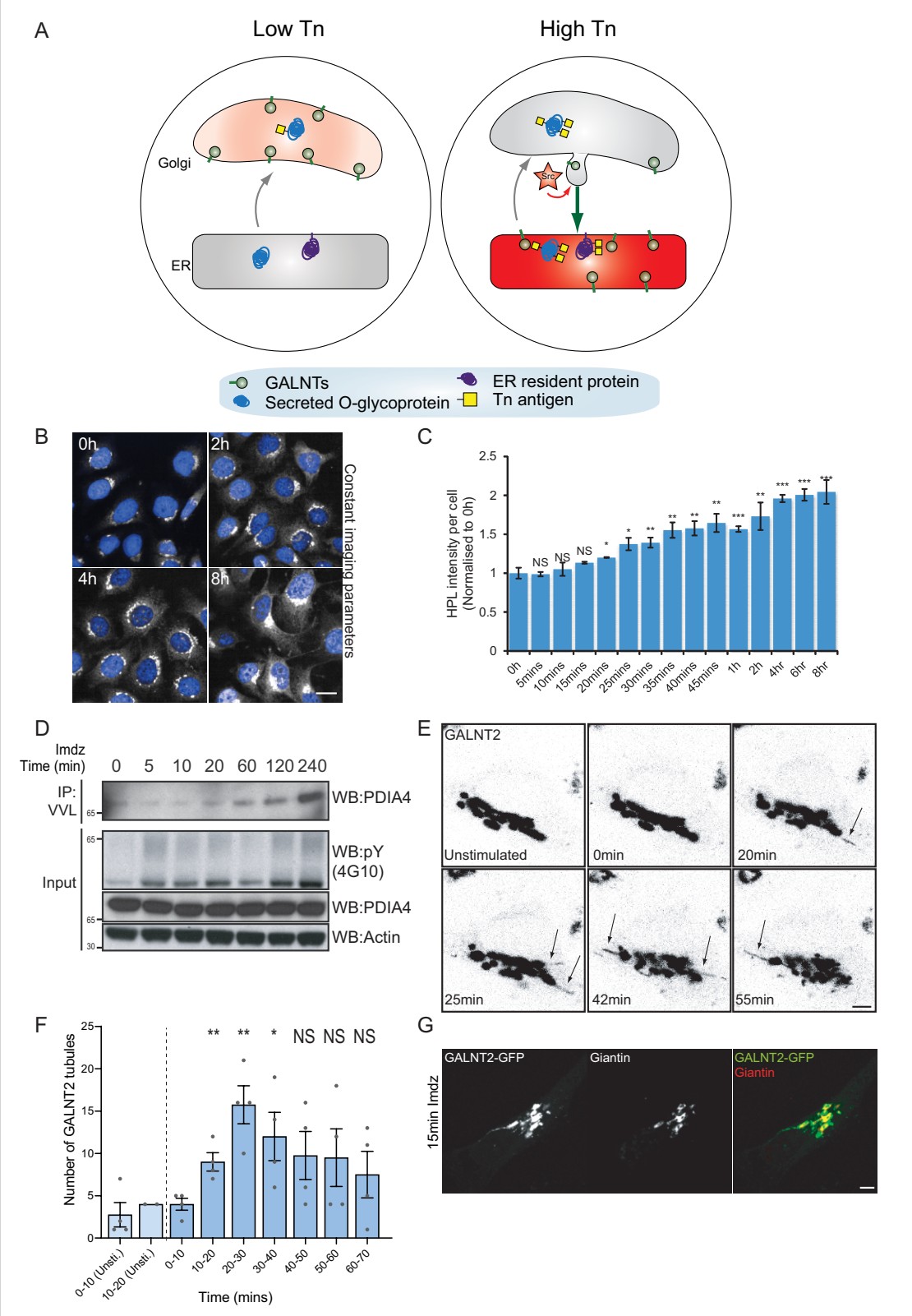

**Figure 1.** Src8A7F chemical activation induces GALNTs relocation to the endoplasmic reticulum (ER) in tubular carriers. (**A**) Schematic of the GALA pathway, the red coloring represents the anti-Tn lectin staining. (**B**) Representative images of *Helix pomatia* lectin (HPL) staining of Tn in HeLa-IS cells after 5 mM imidazole (imdz) stimulation. Scale bar: 20 μm. (**C**) HPL staining intensity per cell normalized to untreated control cells (0 hr). Three replicate wells per experiment were quantified. (**D**) Representative immunoblot analysis of *Vicia villosa* lectin (VVL) immunoprecipitation of cell lysate after 5 mM

*Figure 1 continued on next page*

*Figure 1 continued*

imdz treatment of HEK-IS cells. (**E**) Still images of time-course analysis of GALNT2-GFP-expressing HeLa-IS cells stimulated with 5 mM imdz. (See *Figure 1—video 1* for time-lapse movie.) Scale bar: 5 μm. (**F**) Quantification of the number of GALNT2 tubules emanating from the Golgi over various time pre-imdz (light blue bars) and post-imdz treatment (dark blue bars). Tubules were counted manually over 10 min windows in four independent cells. (**G**) Fixed GALNT2-expressing HeLa-IS cells were stained for the Golgin Giantin. Values on graphs indicate the mean ± SD. Statistical significance (p) was measured by two-tailed paired *t*-test. *p<0.05, **p<0.01, ***p<0.001 relative to untreated cells.

The online version of this article includes the following video and figure supplement(s) for figure 1:

**Figure supplement 1.** Src activation promotes GALNT tubules at the Golgi.

**Figure 1—video 1.** Video of tubule formation in GALNT2-GFP-expressing HeLa-IS cells stimulated with 5 mM imidazole (imdz).

https://elifesciences.org/articles/68678/figures#fig1video1

Src. However, this approach implies an uncontrolled increase of Src activity over several hours and tends to result in fragmentation of the Golgi apparatus (*Bard et al., 2003*).

We sought to obtain a better kinetic control of the Src-induced GALNT relocation. The Cole group has demonstrated that a mutant form of Src, Src(R388A,Y527F) or Src8A7F for short, is a mostly inactive kinase that can be chemically rescued and activated by imidazole (*Qiao et al., 2006*; *Figure 1—figure supplement 1A*). After generating a stable HeLa cell line expressing this *I*nducible *S*rc, HeLa-IS, we verified that imidazole treatment induces an increase in total tyrosine phosphorylation. By comparison with the overexpression of constitutively active point mutant SrcE378G (SrcEG), the increase in tyrosine phosphorylation remained moderate (*Figure 1—figure supplement 1B*).

In terms of GALA, imidazole treatment induced a twofold increase in total Tn levels after 2 hr with a pattern of Tn staining a mix of Golgi and ER, suggesting a measured GALNT relocation (*Figure 1B and C*, *Figure 1—figure supplement 1C*). Staining pattern of the Golgi marker Giantin was not affected, indicating that the Golgi organization was not overly perturbed. Tn increase was relatively modest compared to the expression of SrcEG (*Figure 1—figure supplement 1D*) or ERK8 depletion in HeLa cells (*Chia et al., 2014*). Imidazole treatment in wild-type HeLa cells had no effect on Tn (*Figure 1—figure supplement 1E*). We also observed that the effects of Src8A7F activation were reversible: imidazole washout after 24 hr of treatment resulted in a significant reduction of Tn levels within 1 hr (*Figure 1—figure supplement 1F and G*).

To evaluate the physiological relevance of the inducible Src activation system, we compared it to a stimulation with platelet-derived growth factor (PDGF). PDGF binding to the PDGF receptor usually results in Src activation (*Thomas and Brugge, 1997*). Stimulation with 50 ng/ml PDGF yielded around a twofold increase in total Tn levels (*Figure 1—figure supplement 1H and I*), similar to that observed with imidazole treatment. Hence, Src8A7F rescue recapitulates the levels of response to growth factor stimulation. The advantage of imidazole rescue is a reliable Src activation, whereas the effect of PDGF stimulation tends to be influenced by cell culture conditions (*Chia et al., 2019*). Similar data were obtained in another cell line HEK293 that stably expresses Src8A7F (HEK-IS). HEK-IS recapitulated the results obtained with HeLa-IS, providing an alternative model for biochemical experiments.

We further verified whether increased Tn levels were due to relocation of GALNTs to the ER. Direct observation of GALNTs in the ER is technically challenging because of the dilution and dispersion factor involved (*Chia et al., 2019*). However, GALNTs presence in the ER results in O-glycosylation of ER resident proteins such as PDIA4, which is more readily quantified (*Chia et al., 2019*; *Nguyen et al., 2017*). We measured the effects of Src8A7F imidazole rescue on the glycosylation of PDIA4 using VVL immunoprecipitation (IP) followed by PDIA4 blotting and quantified a fivefold increase of PDIA4 glycosylation after 4 hr (*Figure 1D*). Overall, the Src8A7F system provides a measured activation of the GALA pathway, without breakdown of the Golgi structure and with tight kinetic control.

## Acute activation of Src induces GALNTs-containing tubules at the Golgi

To visualize GALNTs relocation, HeLa-IS cells stably expressing GALNT2-GFP were imaged by time-lapse microscopy after imidazole stimulation. In unstimulated conditions, GALNT2 was mostly confined at the Golgi. Upon imidazole addition, GFP-positive tubules started to emanate from the Golgi as soon as 10 min after stimulation, their numbers reaching peak around 20–30 min, then decreasing to slightly above unstimulated conditions (*Figure 1E and F*, *Figure 1—video 1*). The tubules often detached and moved away from the Golgi, suggesting effective transport (*Figure 1F*).

This phenomenon was also observed after PDGF stimulation where GALNT2 tubules emerged from the Golgi after ~15 min (*Figure 1—figure supplement 1J*). In some particularly responsive cells, the tubules were forming at a high rate and eventually led to a marked reduction of GALNT2 levels in the Golgi. Of note, similar tubules were also observed upon drug inhibition of ERK8, a negative inhibitor of GALNTs relocation (*Chia et al., 2014*). The tubules were deprived of the peripheral Golgi protein Giantin (*Figure 1G*). In addition, they appeared deprived of the chimeric Golgi enzyme beta 1,4-galactosyltransferase (GALT) tagged with mCherry (*Figure 1—figure supplement 1K*).

## Src activation does not increase COPI recruitment on Golgi membranes

We next wondered whether tubule formation was dependent on the COPI coat. We first measured if COPI was recruited at the Golgi upon Src activation using staining for the beta-subunit of COPI. Surprisingly, using both high-resolution microscopy and quantitative automated high-throughput confocal microscopy, we observed no increase but instead a mild reduction of COPI intensity at the Golgi between 5 and 20 min after Src activation (*Figure 1—figure supplement 1L–N*). These results suggest that COPI is not playing a driving role in the formation of tubules at the Golgi, consistent with previous reports about the formation of retrograde-directed tubular intermediates at the Golgi (*Bottanelli et al., 2017*). Since COPI vesicles formation cannot be readily observed, our observations suggest that GALNTs retrograde traffic to the ER is mediated instead by tubules emanating from the Golgi and seceding into transport carriers.

## Transient Src activation increases Arf1-GTP levels

Tubular carriers involved in retrograde traffic have been described previously and recently shown to depend on the small GTPase Arf1 (*Bottanelli et al., 2017*). We previously reported the requirement of Arf1 for GALNTs relocation (*Chia et al., 2014*; *Gill et al., 2010*). By contrast, the small GTPase Arf3 is thought to act primarily in anterograde traffic at the TGN (*Sztul et al., 2019*). Consistently, siRNA knockdown of Arf1 resulted in a significant reduction of Tn levels upon imidazole treatment and Arf3 knockdown had little effect (*Figure 2A and B*, *Figure 2—figure supplement 1E*).

Arf1 has been involved in the formation of retrograde tubular carriers at the Golgi (*Beck et al., 2008*; *Bottanelli et al., 2017*; *Krauss et al., 2008*). We wondered if Arf1 was present on GALNT2 tubules; however, antibody staining was too faint to be conclusive. We thus generated HeLa-IS stably coexpressing GALNT2-GFP and C-terminal V5-tagged Arf1 (Arf1-V5). The small V5 tag was selected to minimize functional interference, and we picked a clone that expresses moderate levels of Arf1-V5 (*Jian et al., 2010*). We found that in unstimulated cells Arf1-V5 localizes both at the Golgi and in peripheral cytosol (*Figure 2C*). Upon Src8A7F activation, Arf1-V5 appeared to be recruited at the Golgi and localized on the GALNT2 tubules, almost throughout the structure (*Figure 2C*, *Figure 2—figure supplement 1A*). To confirm the membrane recruitment of Arf1, we isolated cytosolic and membrane proteins to measure the levels of membrane-bound Arf1 and found Arf1 membrane-association increased within 5 min, peaked at 10 min, and began to fall after 20 min while the cytosolic pool remained relatively constant (*Figure 2D*, *Figure 2—figure supplement 1D*).

The results so far suggested that Arf1 is activated by Src, suggesting an effect on Arf1-GTP levels. We measured them using pulldown with the binding domain of the Arf1 effector GGA1 in HEK-IS (*Dell'Angelica et al., 2000*; *Yoon et al., 2005*). Strikingly, Arf1-GTP levels increased more than twofold within 5 min of imidazole induction (*Figure 2E and F*). Interestingly, Arf1-GTP levels subsided after 30 min of stimulation despite continuous Src activity. Surprisingly, transient expression of SrcEG for 18 hr resulted in a marked decrease in the amount of Arf1-GTP (*Figure 2—figure supplement 1B and C*).

Altogether, the data indicate that Src activation at the Golgi results in a transient increase in GTP-loaded Arf1 and recruitment at the Golgi. Given the reported increased affinity of Arf-GTP for membranes, the switch to GTP-bound form might explain the increase in membrane-bound Arf (*Nawrotek et al., 2016*; *Pasqualato et al., 2002*).

## GBF1 is required for Arf-GTP formation, GALNT relocation, and tubule formation

GTP loading of Arf1 at the Golgi is regulated by GBF (*Kawamoto et al., 2002*; *Zhao et al., 2006*). We previously reported that lowering GBF1 levels reduces GALNTs relocation in cells where GALA

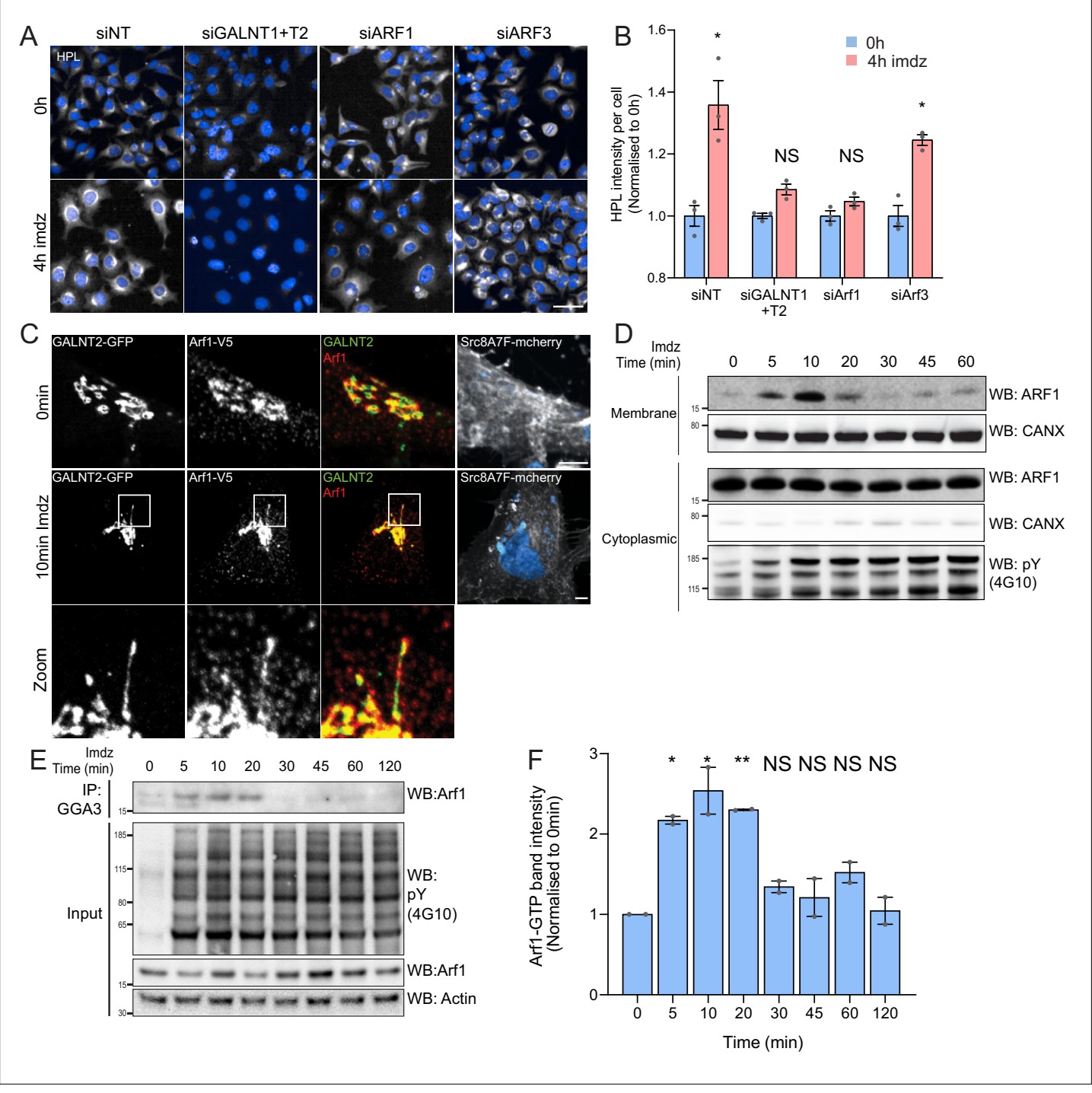

**Figure 2.** Src activation stimulates GTP loading and membrane recruitment of Arf1. (**A**) Representative images of *Helix pomatia* lectin (HPL) staining in HeLa-IS treated with various siRNA before and after 4 hr of imidazole (imdz) treatment. siNT refers to non-targeting siRNA, and siGALNT1 + T2 refers to co-transfection of GALNT1 and GALNT2 siRNAs. Images were acquired under constant acquisition settings. Scale bar: 50 μm. (**B**) Quantification of HPL staining intensity per cell normalized to the respective untreated cells (0 hr) for each siRNA treatment. Three replicate wells per experiment were measured. (**C**) Representative images of GALNT2-expressing HeLa-IS cells stained for Arf1 before and after 10 min of stimulation with 5 mM imdz. Images were acquired at ×100 magnification. Scale bar: 5 μm. (**D**) SDS-PAGE analysis of cytoplasmic and membrane levels of Arf1 after imdz stimulation. CANX refers to blotting for endoplasmic reticulum (ER)-resident Calnexin. The blots were generated with the same exposure and repeated twice. (**E**) SDS-PAGE analysis of GTP-loaded Arf1 after pulldown with GGA3 beads after imdz treatment in HEK-IS cells. (**F**) Quantification of Arf1-GTP levels in (**E**). Two experimental replicates were measured and values were normalized to untreated cells (0 hr). Values on graphs indicate the mean ± SD. Statistical significance (p) was measured by two-tailed paired *t*-test. *p<0.05, **p<0.01, ***p<0.001 relative to untreated cells. NS, nonsignificant.

*Figure 2 continued on next page*

Figure 2 continued

The online version of this article includes the following figure supplement(s) for figure 2:

**Figure supplement 1.** Active Src transiently stimulates nucleotide exchange and membrane recruitment of Arf.

has been induced by ERK8 depletion (*Chia et al., 2014*). Following Src activation in imidazole-treated HEK-IS, GBF1 RNAi-mediated knockdown resulted similarly in a significant reduction of Tn levels (*Figure 3—figure supplement 1A and B*) and PDIA4 glycosylation (*Figure 3—figure supplement 1*). These results indicate that GBF1 is mediating the burst of GALNTs relocation induced by Src8A7F and suggest that GBF1 may also control Arf-GTP burst.

To test this, we reasoned that increased GBF1 expression, together with Src activation, should enhance Arf-GTP formation. Expression of a GFP-tagged form of GBF1 (GFP-GBF1) alone enhances Arf1-GTP levels in HEK-IS (*Figure 3A*). Strikingly, upon imidazole stimulation, GTP loading was further increased by nearly threefold within 10 min of induction (*Figure 3A and B*). The effect was transient and Arf1-GTP returned to pre-stimulation within 45 min, indicating similar dynamics as with wild-type levels of GBF1.

We next tested the effect of GBF1 depletion on Arf-GTP production upon Src activation. GBF1 knockdown affects cell adhesion and Golgi morphology, but these effects occur progressively after siRNA transfection. To capture recently GBF1-depleted cells, we harvested them at 48 hr instead of 72 hr. In these conditions, GBF1 levels were already significantly lowered; imidazole stimulation did not result in significant Arf-GTP production (*Figure 3C*). Averaging three independent experiments resulted in over 90% reduction of Arf activation (*Figure 3D*).

Since GBF1 is involved in Arf-GTP production after Src activation, we reasoned it might be recruited to Golgi membranes. Strikingly, the GBF1 membrane pool in HEK-IS increased by roughly threefold within 5–10 min, while total GBF1 remained constant. This increase was partially transient, peaking at 10 min of imidazole treatment, but remained elevated for up to 60 min (*Figure 3E and F*). Using time-lapse microscopy of GFP-GBF1 in HeLa-IS, we observed GBF1 recruitment at the Golgi complex. In sync with the biochemical experiment, GBF1 Golgi levels increased rapidly, peaking at ~10 min, decreasing after ~20 min but not to baseline levels (*Figure 3G and H*).

GBF1 recruitment at the Golgi depends on binding to membrane-bound Arf-GDP (*Quilty et al., 2014*). Thus, we wondered whether phosphorylation by Src might increase GBF1 affinity for Arf1. To test this idea, we isolated Arf1-V5 from a cell lysate and added, in the presence of GDP, GFP-GBF1 immunoprecipitated on beads from cells expressing also active or inactive Src (*Figure 3—figure supplement 1D*). After 1 hr incubation, beads were washed and the amount of Arf1 bound to beads quantified by immunoblotting (*Figure 3I*). By comparison with inactive SrcKM, SrcEG induced a twofold increase in binding (*Figure 3—figure supplement 1E*). Given this net increase, we next tested binding of immunoprecipitated GFP-GBF1 with bacterially produced and purified Arf1-del17-His in the presence of GDP. Arf1-del17-His, a recombinant protein deleted of the first 17 amino acids, is able to bind GDP in the absence of phospholipids (*Kahn et al., 1992*). As with Arf1-V5, phosphorylated GBF1 displayed increased binding to purified Arf1-del17-His (*Figure 3—figure supplement 1F*).

We then repeated the imaging of GALNT2-GFP in live HeLa-IS cells with cells depleted for GBF1. After imidazole stimulation, control cells displayed abundant tubule formation while tubules were almost nonexistent in GBF1-depleted cells (*Figure 3—figure supplement 1G and H*, *Figure 3—videos 1 and 2*).

Altogether, our results indicate that Src activation induces a wave of Arf-GTP that appears driven by an increase of affinity between GBF1 and Arf1-GDP. Coincidently, GBF1 is recruited to Golgi membranes and is involved in the formation of GALNT2 tubules.

## GBF1 protein is phosphorylated by Src on at least 10 tyrosine residues

We next tested directly whether Src phosphorylates GBF1. After imidazole stimulation of HEK-IS cells, GBF1 was immunoprecipitated and probed with an antibody specific for tyrosine phosphorylation, revealing an increase within 5 min that was sustained for 2 hr (*Figure 4A*). As the signals both for GBF1 and its phosphorylation were not very marked, we also tested GFP-GBF1-expressing HEK-IS cells, obtaining similar results (*Figure 4B*). We also transiently coexpressed GFP-GBF1 with SrcEG and SrcKM mutants in HEK293T cells. In such conditions, the difference in phosphorylation levels was very marked (*Figure 4C*). Similarly, endogenous GBF1 was phosphorylated by SrcEG in

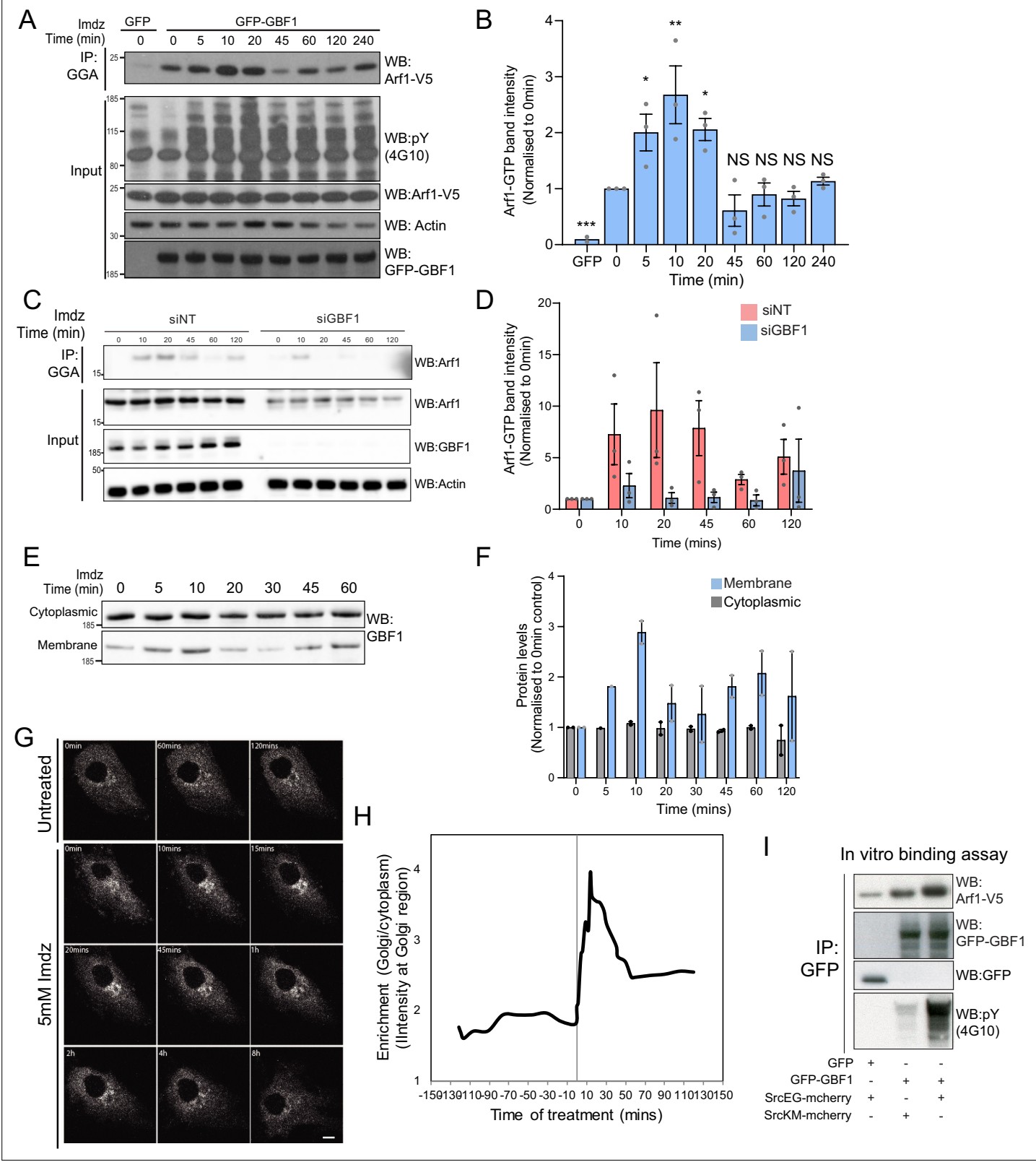

**Figure 3.** Src activates the ARF-GEF GBF1. (**A**) Representative SDS-PAGE analysis of Arf1-GTP levels in HEK-IS cells expressing GFP or GFP-GBF1. GGA pulldown was performed as in *Figure 2E*. (**B**) Quantification of Arf1-GTP levels in three independent experiments in (**A**). (**C**) SDS-PAGE analysis of Arf1-GTP levels in HEK-IS cells treated with siGBF1 and siNT siRNA. (**D**) Quantification of the Arf1-GTP levels in three independent experiments in (**C**). (**E**) SDS-PAGE analysis of cytoplasmic and membrane levels of GBF1 after imidazole (imdz) stimulation. (**F**) Quantification of two independent

*Figure 3 continued on next page*

*Figure 3 continued*

experiments shown in (**E**). Values presented were normalized to untreated cells (0 hr).(**G**) Still images of the time-lapse movie of GBF1-GFP in HeLa-IS cells stimulated with 5 mM imdz. Scale bar: 10 μm. (**H**) Quantification of the ratio of Golgi to total cytoplasmic levels of GBF1 before and after imdz treatment in time lapse shown in (**G**). (**I**) SDS-PAGE analysis of the levels of Arf1-V5 bound to GFP or GFP-GBF1 immunoprecipitation (IP) from cells expressing inactive SrcKM or active SrcEG in an in vitro binding assay. Two experimental replicates were tested and quantified in *Figure 3—figure supplement 1E*.

The online version of this article includes the following video and figure supplement(s) for figure 3:

**Figure supplement 1.** GBF1 is required for GALNT tubule formation.

**Figure 3—video 1.** Video of tubule formation in siNT-treated HeLa-IS cells expressing GALNT2-GFP stimulated with 5 mM imidazole (imdz).
https://elifesciences.org/articles/68678/figures#fig3video1

**Figure 3—video 2.** Video of tubule formation in siGBF1-treated HeLa-IS cells expressing GALNT2-GFP stimulated with 5 mM imidazole (imdz).
https://elifesciences.org/articles/68678/figures#fig3video2

HeLa cells (*Figure 4—figure supplement 1A*). Finally, we tested GBF1 phosphorylation in a third cell line: NIH3T3vSrc are mouse fibroblasts that have transformed with a viral, oncogenic, and constitutively active mutant of Src (*Vogt, 2012*). These cells display significantly higher levels of GALA than their normal counterparts (*Figure 4—figure supplement 1B*). They also display four times as much phospho-GBF1 (*Figure 4D*). To test whether phosphorylation was direct, we immunoisolated GFP-GBF1 from HEK293 cells and added recombinant Src. In the presence of ATP, GFP-GBF1 displayed marked phosphotyrosine levels compared to controls, indicating that GBF1 is a direct substrate of Src (*Figure 4E*, *Figure 4—figure supplement 1C*).

To map the phosphorylation sites, GFP-GBF1 from HEK293 cells expressing active SrcEG was extracted from a gel separation and digested using trypsin and AspN. Phosphopeptides were analyzed with tandem mass spectrometry, revealing 10 phosphorylated tyrosine residues with high confidence (*Figure 4F*, *Figure 4—figure supplement 1D*). All the phosphopeptides identified were clearly increased in the presence of SrcEG but mostly not detectable with inactive SrcKM expression (*Figure 4—figure supplement 1D*).

## Src phosphorylates two tyrosines at the C-terminus of GBF1 GEF domain

Ten phosphosites are challenging to study in parallel. We were particularly interested in the residues Y876 and Y898 because they are located respectively within the GEF/Sec7d and on a C-terminal loop connecting Sec7d and the HDS1 domain (*Figure 4F*). It suggested that they could be directly involved in the regulation of GBF1 GEF activity.

A database search revealed that Y876 is conserved in GBF1 homologues in various species (*Figure 4G*). Y898 is also conserved in GBF1 in most species (*Figure 4G*, *Figure 4—figure supplement 1E*). In contrast, the other phosphorylation sites are mostly conserved in vertebrates and Y317 is only present in human GBF1 (*Figure 4—figure supplement 1E*). We next compared Sec7 domains of different ARFGEFs and found Y876 to be highly conserved, while Y898 appeared specific for GBF1. Interestingly, a search on PhosphoSitePlus database indicates that multiple GEFs, including Brefeldin-Resistant Arf-GEF 2 (BRAG2, IQSEC1), Brefeldin A-Inhibited Guanine Nucleotide-Exchange Protein 1 (BIG1), and Cytohesin 2 (ARNO), can be phosphorylated on the tyrosine analogous to Y876 (*Figure 4H*).

We subsequently used targeted SILAC to quantify the intensity of phosphorylation on each site in HEK293 transiently transfected (*Ong et al., 2002*). GBF1, in the presence of SrcEG, displayed about 180-fold increase in phosphorylated Y876 peptide (DFEQDILEDMyHAIK) and 100-fold in increase in phosphorylated Y898 peptide (ENyVWNVLLHR) compared to SrcKM-expressing samples, suggesting that these sites are the major phosphorylation targets (*Figure 4—figure supplement 1F–H*).

## Phosphorylation at Y876 in endogenous GBF1 is confirmed with a specific antibody

We next aimed to generate antibodies specific for phospho-Y876 and Y898. While our efforts on Y898 were unsuccessful, we obtained a monoclonal antibody named 2P4 after immunization with a Y876-containing phosphopeptide that reacted with phospho-GBF1 (*Figure 4—figure supplement 1I*). To

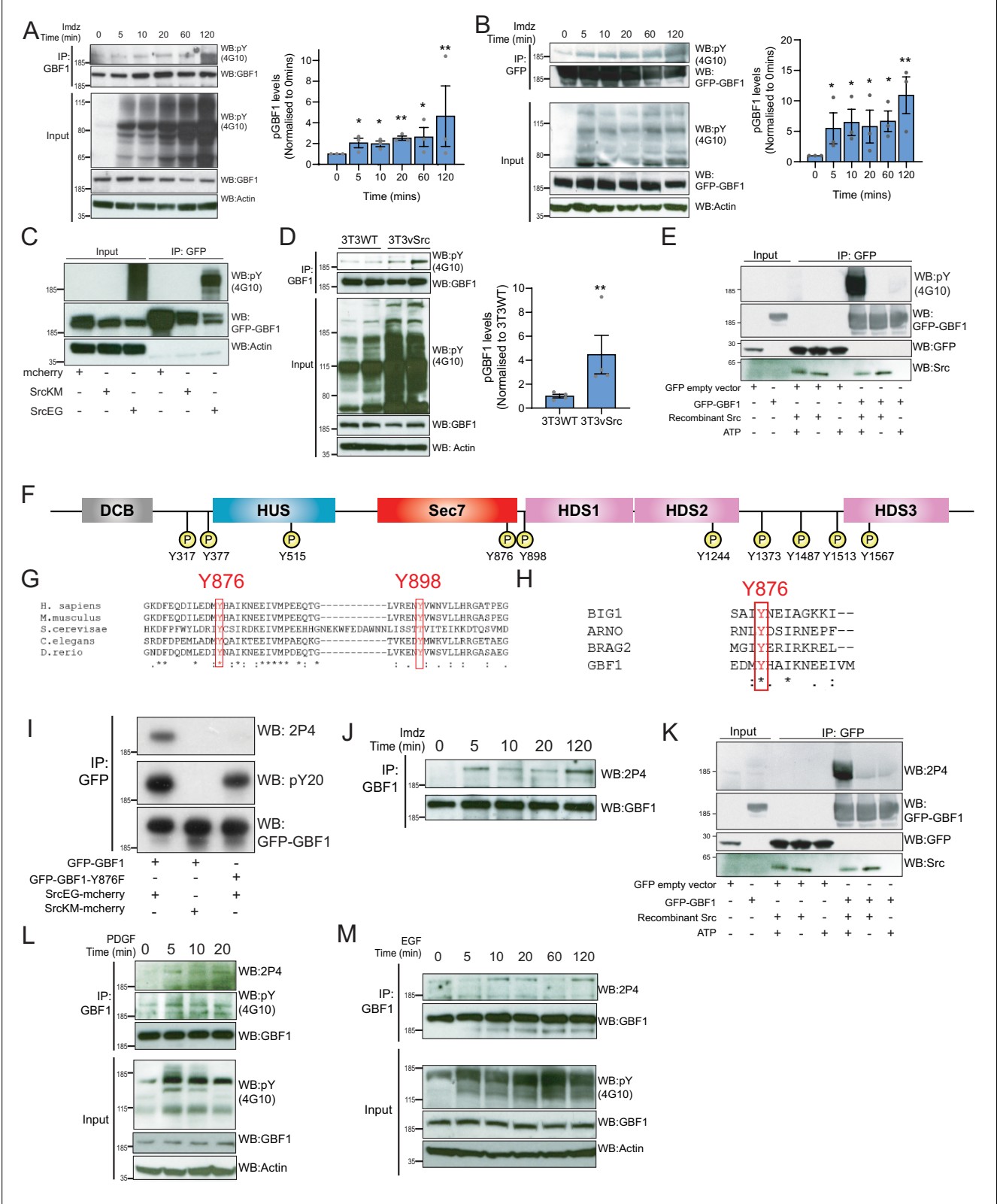

**Figure 4.** Src phosphorylates two tyrosines Y876 and Y898 at the C-terminus of the GBF1 Sec7d. (**A**) SDS-PAGE analysis of phosphotyrosine (pY) levels in endogenous GBF1 using HEK-IS cells after imidazole (imdz) treatment. Quantification of pY-GBF1 in three replicates shown on the graph (right). (**B**) SDS-PAGE analysis of pY levels on GFP-GBF1 immunoprecipitation (IP) from HEK-IS cell line after imdz treatment. Quantification of pY-GBF1 in three replicates shown on the graph (right). (**C**) SDS-PAGE analysis of pY in GBF1 in HEK293T cells expressing either inactive SrcKM or active SrcEG. (**D**) SDS-

*Figure 4 continued*

PAGE analysis of pY levels on endogenous GBF1 IP from wild-type and vSrc transformed NIH3T3 cell lines. Quantification of pY-GBF1 in four replicates from two independent experiments shown on the graph (right). (**E**) Quantification of pY in GBF1 after in vitro phosphorylation. Immunoprecipitated GFP or GFP-GBF1 was incubated with recombinant Src protein in the presence or absence of nucleotide ATP. (**F**) Schematic of the 10 tyrosine residues in GBF1 that were identified by targeted mass spectrometry after exposure to Src. (**G**) Amino acid sequence alignment of GBF1 from various species. The sequences of GBF1 at Y876 and Y898 of *Homo sapiens* (NP_004184) was aligned with that of *Mus musculus* (NP_849261), *Saccharomyces cerevisiae* (NP_010892), *Caenorhabditis elegans* (NP_001255140), and *Danio rerio* (XP_009305378), revealing conservation of both residues. (**H**) Y876 is conserved and observed to be phosphorylated in other GEFs BRAG2, ARNO, and BIG1 based on the PhosphoSitePlus database. (**I**) SDS-PAGE analysis of wild-type GFP-GBF1 or GFP-GBF1-Y876F mutant immunoprecipitated from HEK293T cells expressing inactive SrcKM or active SrcEG. Phosphorylation at Y876 was marked by the 2P4 antibody. (**J**) SDS-PAGE analysis of Y876 phosphorylation on endogenous GBF1 IP from HEK-IS cell line over various durations of imdz treatment. (**K**) Y876 phosphorylation of GBF1 in an in vitro phosphorylation assay. (**L**) SDS-PAGE analysis of the total and Y876 phosphorylation on endogenous GBF1 in HeLa cells over the duration of 50 ng/ml platelet-derived growth factor (PDGF) stimulation. (**M**) SDS-PAGE analysis of Y876 phosphorylation on endogenous GBF1 in A431 cells over time of 100 ng/ml EGF stimulation. Values on graphs indicate the mean ± SD. Statistical significance (p) was measured by two-tailed paired *t*-test. *p<0.05, **p<0.01, ***p<0.001 relative to untreated cells. NS, nonsignificant.

The online version of this article includes the following figure supplement(s) for figure 4:

**Figure supplement 1.** High cross-species conservation of tyrosine residues phosphorlated in GBF1.

verify specificity, wild-type GFP-GBF1 and mutant GFP-GBF1(Y876F) were coexpressed with SrcEG, immunoprecipitated, and probed with 2P4. While wild-type GBF1 showed strong reactivity and total phosphotyrosine levels were moderately affected, the band was completely abolished in the Y876F mutant (*Figure 4I*).

We used 2P4 to assess the kinetics of Y876 phosphorylation after Src activation in the HEK-IS system. Similarly to overall phosphotyrosine levels, phospho-Y876 was detected within 5 min of imidazole treatment and persisted for 2 hr (*Figure 4J*). Similar results were obtained with GFP-GBF1 (*Figure 4—figure supplement 1J*). We also verified that Y876 is a direct target of Src using the in vitro phosphorylation assay (*Figure 4K*, *Figure 4—figure supplement 1K*).

We also tested Y876 phosphorylation after growth factor stimulation. Starting with serum-starved HeLa cells stimulated with PDGF, endogenous GBF1 was immunoprecipitated and phospho-Y876 found to display similar kinetics to generic GBF1 tyrosine phosphorylation (*Figure 4L*). A431 cells, which express high levels of EGFR, were stimulated with 100 ng/ml of EGF (*Fernandez-Pol, 1985*). Similarly to PDGF with HeLa cells, phospho-Y876 was upregulated within 10–20 min (*Figure 4M*). To review, Y876 is a major site of phosphorylation by Src and is modified in physiological conditions of GALA activation.

## Phosphorylation at Y876 and Y898 is required for Src-induced Arf1-GTP levels and GALNT relocation

To test the functional importance of Y876 and Y898, we generated single and double tyrosine (Y) to phenylalanine (F) phospho-null mutants at position Y876 and Y898 (Y876F, Y898F, Y876.898F). We then measured Arf1 GTP loading after Src8A7F activation in the presence of transiently expressed wild-type or phospho-defective mutant GBF1. As previously observed, Arf1-GTP increased by ~2.5-fold within 10 min of imidazole treatment with wild-type GBF1 expression. This increase was abolished with the Y876F and double mutant, while residual activation was observed with the Y898F mutant (*Figure 5A and B*). Expression of the double mutant even reduced the basal Arf1-GTP levels.

Next, we tested whether GALNT relocation was affected by the phospho-defective mutants by measuring Tn levels. GBF1-Y876F or GBF1-Y898F significantly repressed Tn levels in SrcEG-expressing cells (*Figure 5C and D*, *Figure 5—figure supplement 1A*). There was also a reduction of Tn levels by phospho-defective GBF1 in the presence of activated Src8A7F (*Figure 5—figure supplement 1B*). These results indicate that the GBF1 mutants do not support the formation of GALNT retrograde transport carriers and may act as dominant-negative.

We next tested their effect on Arf binding. In the in vitro binding assay with purified Arf1-del17 and immuno-purified GFP-GBF1 wild-type and phospho-defective proteins, single mutants of Y876 or Y898 had reduced Arf1 binding by 40 and 60%, respectively, with more than 70% for the double mutant (*Figure 5E and F*). In fact, the double mutant had lower binding to Arf than nonphosphorylated wild-type GBF1. These results confirm that the phosphorylation on tyrosines Y876 and Y898 increases affinity of GBF1 for Arf1-GDP.

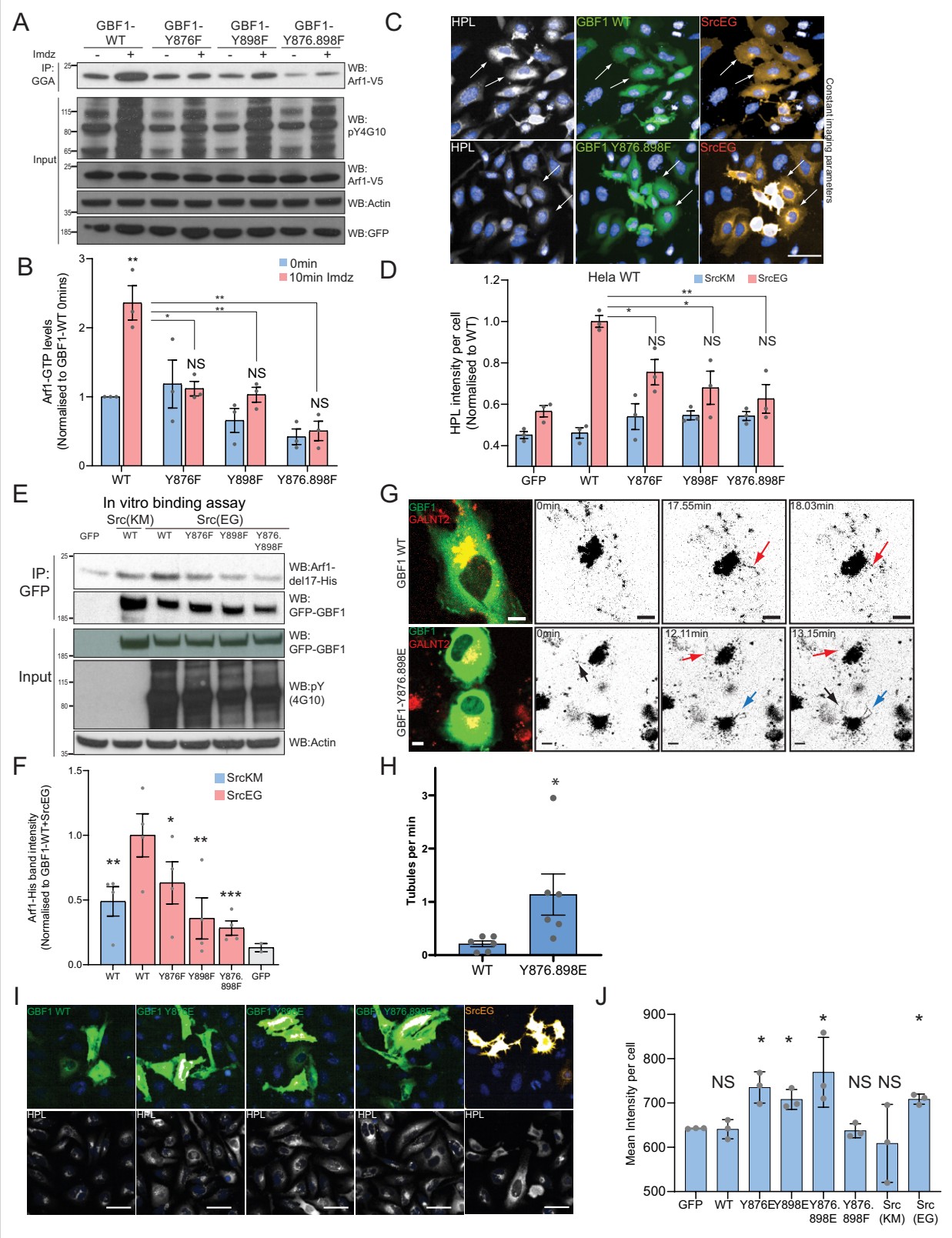

**Figure 5.** Phosphorylation at Y876 and Y898 regulates GEF activity of GBF1. (**A**) SDS-PAGE analysis of GTP-loaded Arf1 at 0 min (-) and 10 min (+) imidazole treatment in HEK-IS cells expressing wild-type GBF1 and phospho-defective mutants GBF1-Y876F, GBF1-Y898F, or GBF1-Y876.898F. (**B**) Quantification of Arf1-GTP loading levels in (**A**) from three independent experiments. Values were normalized to untreated cells (-) expressing wild-type GBF1. (**C**) Representative images of *Helix pomatia* lectin (HPL) staining in HeLa cells coexpressing wild-type GBF1 or GBF1-Y876.898F

*Figure 5 continued*

mutant with active SrcEG. Scale bar: 50 µm. (**D**) Quantification of HPL staining levels of cells coexpressing wild-type or mutant GBF1 with inactive SrcKM (blue bars) or active SrcEG (pink bars). Values were from three replicates. (**E**) SDS-PAGE analysis of the levels of recombinant Arf1-His bound to wild-type or phospho-defective GBF1 mutants immunoprecipitated (IP) from cells expressing inactive SrcKM or active SrcEG in an in vitro binding assay. (**F**) Quantification of the levels of bound Arf1-His. Values were from four experimental replicates and normalized to wild-type GBF1 IP from cells expressing active SrcEG from each experiment. IP'ed GFP protein used as a negative control for nonspecific binding with GFP (gray bar). (**G**) Still images from time-lapse imaging of GALNT2-mCherry in HeLa cells that were either transfected with wild-type GBF1 (GBF1 WT) or phosphomimetic mutant (GBF1 Y876.898E). Arrows indicate tubule formation. Scale bar: 5 µm. (**H**) Quantification of the number of tubules per minute of acquisition. (**I**) Representative images of HPL staining in HeLa cells expressing phosphomimetic (Y-to-E) mutants 4 hr post transfection. (**J**) Quantification of HPL staining levels in (**I**). Values on graphs indicate the mean ± SD. Statistical significance (p) was measured by two-tailed paired *t*-test. *p<0.05, **p<0.01, ***p<0.001 relative to untreated cells or to 10 min imidazole (imdz)-treated cells expressing wild-type GBF1. NS, nonsignificant.

The online version of this article includes the following video and figure supplement(s) for figure 5:

**Figure supplement 1.** Phosphomimetic mutants at Y876 and Y898 recapitulate the effects of active Src.

**Figure 5—video 1.** Video of tubule formation in GALNT2-mCherry-expressing HeLa cells transfected with wild-type GBF (GBF1 WT).
https://elifesciences.org/articles/68678/figures#fig5video1

**Figure 5—video 2.** Video of tubule formation in GALNT2-mCherry-expressing HeLa cells transfected with phosphomimetic GBF1 mutant (GBF1 Y876.898E).
https://elifesciences.org/articles/68678/figures#fig5video2

## A phosphomimetic mutant at Y876 and Y898 recapitulates GALNT tubule formation

Next, we tested the effect of Y to E phosphomimetic mutations (Y876E and Y898E). First, we observed that after 24 hr overexpression, we observed a reduction of basal GTP loading levels by more than 70% (*Figure 5—figure supplement 1C and D*). While it was initially surprising, this result actually recapitulates the effect of long-term expression of constitutively active SrcEG (*Figure 2—figure supplement 1B*). It suggests that the Y-to-E mutants may induce the same effects with similar kinetics on Arf nucleotide loading than Src. Similarly to Src expression, long-term (~48 hr) expression of the phosphomimetic GBF1 mutant reduces COPI levels at the Golgi.

We next tested if phosphomimetic GBF1 mutant could induce the relocation from the Golgi of GALNTs. We imaged GALNT2-mCherry in cells very transiently expressing phosphomimetic double mutant GFP-GBF1 Y876.898E (YE). We performed this experiment at early times (12–24 hr) after transfection when the GFP becomes just detectable. Remarkably, cells expressing the YE mutant showed a high incidence of tubule formation, while cells expressing wild-type GBF1 (GBF1 WT) displayed very little (*Figure 5G and H*, *Figure 5—figure supplement 1E and F*, *Figure 5—videos 1 and 2*). In YE mutant-expressing cells, tubules appeared also to be more extended and to persist longer than with WT GBF1 (*Figure 5G*, *Figure 5—figure supplement 1E*; red and blue arrows). YE mutant cells displayed a fivefold higher rate of tubule formation per minute of imaging compared to their WT counterpart (*Figure 5H*).

Expression of the phosphomimetic mutants of GBF1 also resulted in a mild but significant increase in Tn levels in cells transfected with YE mutants compared to GBF1 WT cells (*Figure 5I and J*), indicating GALNT relocation to the ER. Altogether, the data indicate that tyrosine phosphorylation of GBF1 is a key driver for GALNT retrograde tubule traffic.

## Phosphorylation on Y898 probably releases a Sec7d-HDS1 intramolecular interaction

We next wondered how the phosphorylations affect GBF1's activity. Y898 is located in the linker region between the Sec7d and HDS1 domains of GBF1. While the Sec7d structure of GBF1 has not been resolved, GBF1 Sec7d shares at least 65% homology with several other GEFs, so it can be modeled relatively accurately. Unfortunately, this is not the case for HDS1 for which there is no structural information. We could model the Sec7d and the linker domain using the GBF1 sequence and the resolved structures of the GEFs ARNO, Cytohesin-1, and Grp1. In this model, the linker is located close to a pocket of negatively charged residues in the Sec7d (*Figure 6—figure supplement 1A*). Molecular dynamics (MD) revealed a repulsion of the linker away from Sec7d after phosphorylation (*Figure 6—figure supplement 1B*, *Figure 6—video 1*). This suggests that phosphorylation could

relieve an intramolecular interaction between the Sec7 domain and the HDS1 domain. ARNO/Grp1/Cytohesin and BIG family contain Pleckstrin Homology (PH) and HDS domains, respectively, in C-term of Sec7d that interacts and inhibits the GEF activity for ARNO by about 14-fold (*Stalder et al., 2011*) and about 7-fold for BIG (*Richardson et al., 2012*). Therefore, we hypothesize that the HDS1 of GBF1 could similarly inhibit the Sec7d and phosphorylation at Y898 would alleviate this Sec7d-HDS1 inhibition.

## Phosphorylation of Y876 partially unfolds GBF1 Sec7d domain, increasing affinity for Arf1

By contrast with Y898, Y876 is located within the Sec7d. There are 10 alpha helices in Sec7d, and Y876 is present on helix J (*Cherfils et al., 1998*). We modeled the interaction of GBF1 with Arf1 based on available structures and observed that helix J is protruding in the interface between the two proteins (*Figure 6—video 2*). When we simulated Y876 phosphorylation, the negative charge was attracted by the positively charged residues arginine 843 (R843) and lysine 844 (K844) at the end of helix H. This, in turn, led to a partial unwinding of helix H, leading to an extension of the loop between helices H and I (*Figure 6A*). This partial unfolding and loop extension would result in better bond formation between Sec7d and Arf1 (*Figure 6B*, *Figure 6—figure supplement 2A*). This translates into a reduced free binding energy (*Figure 6C*, *Figure 6—figure supplement 2B*) and an increased probability of buried surface area (BSA) between the Sec7d and Arf1 (*Figure 6D*) in the phosphorylated state. Higher BSA indicates tighter packing interactions, and thus, higher affinity between the molecules. Thus, the loop extension induced by phosphorylation is predicted to favor the interaction with Arf1, a result consistent with our in vitro binding assay results with phospho-GBF1 (*Figures 3I and 5E*).

Since the model predicts that the positive charges on either R843 and K844 are important for the conformational change induced by phosphorylation, we mutated these sites into glutamic acids (R843E.K844E) or neutral charges with alanine (R843A.K844A). We next tested if these HI loop mutants have an impact on GTP loading on Arf1 in cells. The introduction of negative charges in the R843E.K844E mutant resulted in a massive reduction in basal cellular Arf1-GTP levels by about 90% (*Figure 6E and F*). On the other hand, the mutant with neutral charges (R843A.K844A) did not have an effect on the basal Arf1-GTP levels. However, when we stimulated Src8A7F with imidazole, the cells expressing the R843A.K844A mutant did not upregulate Arf1 GTP loading (*Figure 6E and F*). These results indicate that the positively charged residues in the loop between helices H and I are required for the Y876 phosphorylation effect.

The model predicts that blocking the partial unfolding of helix H would reduce the interaction of GBF1 with Arf1 (*Figure 6B*, *Figure 6—figure supplement 2A*). As expected, we found that the mutants (both E and A) were insensitive to SrcEG in terms of enhanced binding to Arf1-GDP (*Figure 6G*).

We next tested whether blocking helix H unfolding would prevent Src-induced GALNT relocation to the ER. HPL staining intensities in cells coexpressing constitutively active SrcEG and wild-type or the HI loop mutants (both A and E forms) were measured. The HI loop mutants resulted in a significant reduction in Tn levels, at levels similar to GBF1-Y876F *Figure 6—figure supplement 2C and D*. A similar reduction was observed in HeLa Src8A7F cells stimulated with imidazole (*Figure 6—figure supplement 2E and F*).

Altogether, these results strongly support a model where the phosphorylation on Y876 induces a partial melting of the Sec7d helix H, which in turn facilitates GBF1 binding to Arf1-GDP.

## Discussion

In this report, we describe how a tyrosine kinase, Src, controls a key regulator of membrane trafficking, GBF1, and in turn mediates the movement of Golgi enzymes. The relocation of the GALNTs from the Golgi to the ER has been nicknamed the GALA pathway, for GALNTs activation, and its importance for tumor growth has been described previously (*Gill et al., 2013*; *Nguyen et al., 2017*; *Ros et al., 2020*). GALA induction by the oncogenic Src kinase has been described, but it remains unclear how Src induces this relocation (*Gill et al., 2010*).

Here, we propose a model where Src phosphorylates GBF1 on multiple tyrosines, which results in an extrusion of GALNTs from Golgi membranes in tubules and eventually their relocation to the ER

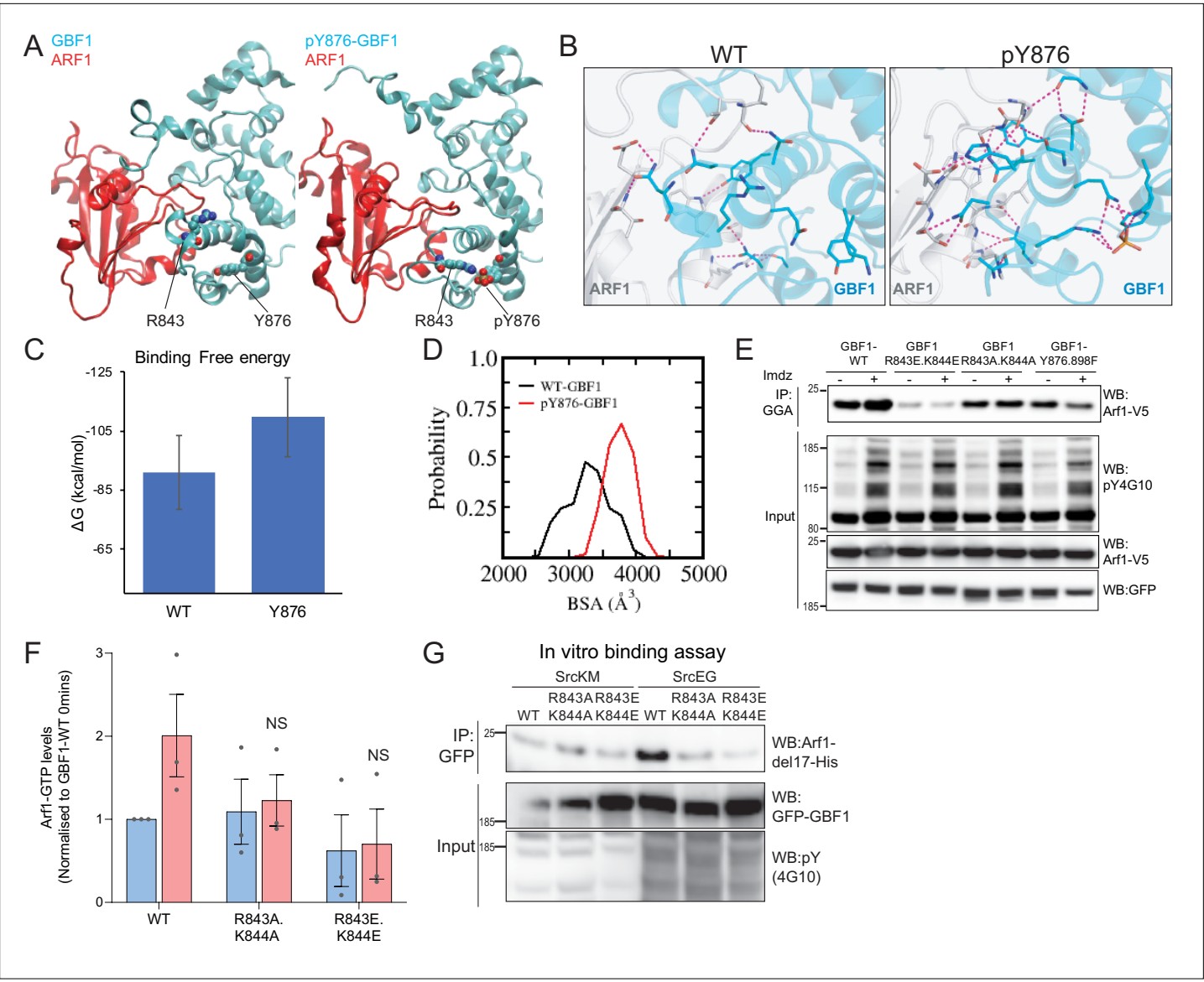

**Figure 6.** Phosphorylation on Y876 increases GBF1 Sec7d affinity for Arf1. (**A**) Structural basis for the binding of unphosphorylated GBF1 Sec7d and Y876-phosphorylated GBF1 Sec7d with Arf1. Cartoon representations of conformations extracted from the molecular dynamics (MD) simulations of the unphosphorylated GBF1 Sec7d (left of panel **A**; cyan) and Y876-phosphorylated GBF1 Sec7d (right of panel **A**; cyan) bound to Arf1 (red color). MD suggests the unwinding of the helix H to form an extended loop between helices H and I through increased attractions between positive charges on R843 and K844 on the loop with the negative charges on phosphorylated Y876 (see *Figure 6—video 2*). The Sec7d of GBF1 (blue), in turn, interacts more with Arf1 (red). (**B**) Residues involved in GBF1:Arf1 interprotein are shown as sticks, and the H-bonds highlighted as black dashed lines. The Sec7d is shown in blue while Arf1 protein is in gray. Refer to *Figure 6—figure supplement 2* for the identities of the residues. (**C**) Estimation of the free energies (ΔG) of the interactions between the unphosphorylated GBF1 Sec7d and Arf1 and between the Y876-phosphorylated GBF1 Sec7d and Arf1. Calculations carried out using the MMPBSA approximations averaged over the conformations generated from MD simulations of the complexes; larger negative values represent higher affinities. GBF1 Sec7d has a higher affinity for Arf1 when it is phosphorylated at Y876. (**D**) Probability distributions of the buried surface area (BSA) between GBF1 Sec7d and Arf1. (**E**) SDS-PAGE analysis of GTP-loaded Arf1 at 0 min (-) and 10 min (+) imidazole treatment in HEK-IS cells expressing wild-type GBF1, Y876.898F, and the HI loop mutants. (**F**) Quantification of Arf1-GTP loading levels in (**E**) in three experimental replicates. Values were normalized to untreated cells (-) expressing wild-type GBF1. (**G**) SDS-PAGE analysis of the levels of recombinant Arf1-His bound to wild-type or the HI loop mutants GFP-GBF1 immunoprecipitation (IP) from cells expressing inactive SrcKM or active SrcEG in an in vitro binding assay. Values on graphs indicate the mean ± SD. Statistical significance (p) was measured by two-tailed paired *t*-test. NS, nonsignificant.

The online version of this article includes the following video and figure supplement(s) for figure 6:

**Figure supplement 1.** Phosphorylation at Y898 appears to release an intramolecular interaction between Sec7d and HDS1 domain.

**Figure supplement 2.** Positive-charged residues R843 and K844 are required for the Y876 phosphorylation effect.

*Figure 6 continued on next page*

*Figure 6 continued*

**Figure 6—video 1.** Molecular dynamics (MD) simulation of the Sec7d and C-terminal linker of wild-type GBF1 (GBF1-WT, left) and when Y898 is phosphorylated (GBF1-pY898, right).
https://elifesciences.org/articles/68678/figures#fig6video1

**Figure 6—video 2.** Molecular dynamics (MD) simulation of unphosphorylated GBF1 Sec7d (GBF1-WT, left panel, cyan) and Y876-phosphorylated GBF1 Sec7d (GBF1-pY876, right panel, cyan) with Arf1 (in red).
https://elifesciences.org/articles/68678/figures#fig6video2

(*Figure 7A*). The two phosphorylation sites at and near the Sec 7 domain of GBF1 increase its affinity for Arf-GDP (*Figure 7B*). Thus, Src increases the rate of retrograde traffic of GALNTs through the phosphorylation of GBF1 and regulation of the GBF1-Arf interaction.

Src kinase activity is critical for its oncogenic activity, and decades of research have revealed that Src is a highly regulated kinase with a large repertoire of substrates (*Reynolds et al., 2014*). GBF1 was not previously recognized as a prominent substrate, and its phosphorylation sites appear only partially in some databases of systematic mass spectrometry approaches. In our study, we found that the GBF1-phosphorylated form is relatively difficult to detect in comparison with other substrates. This is partly because GBF1 protein levels are relatively low, almost 50 times less abundant than Arf1 (*Itzhak et al., 2016*). In addition, others have reported that GBF1 is an unstable protein, which we have also observed with recombinant GBF1 (*Bhatt et al., 2016*). These characteristics might be evolutionary linked to the 'toxicity' of GBF1, whose overexpression affects Golgi organization and cell adhesion. While being a quantitatively minor Src substrate, GBF1 might be a critical substrate for the oncogenic

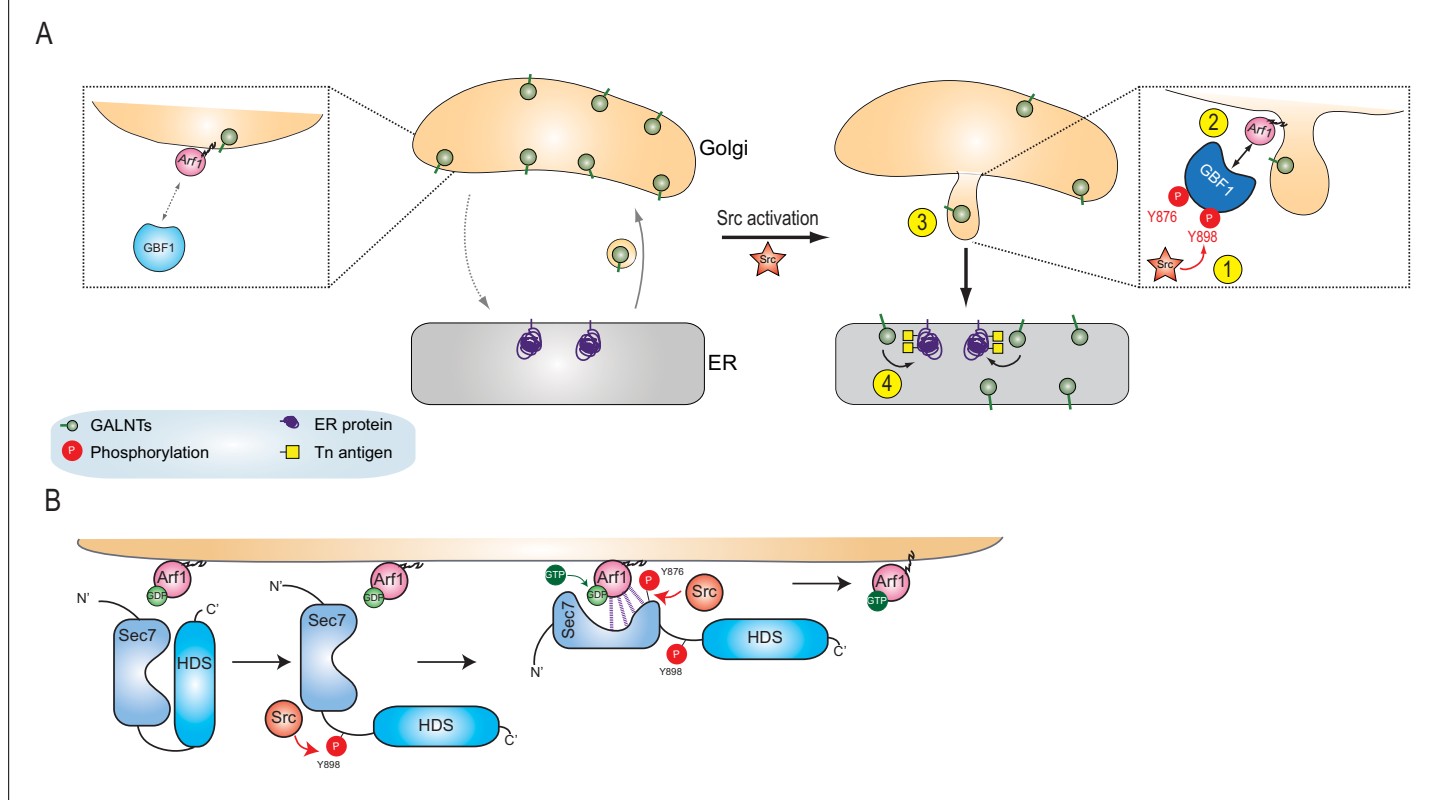

**Figure 7.** Models of GBF1 phosphorylation effects on binding to Arf1 and on the formation of tubules. (**A**) In normal cells, GBF1 and Arf interaction are limited and retrograde traffic rate is moderate. Upon Src activation, GBF1 is phosphorylated on at least two tyrosines, Y876 and Y898 (step 1), which results in increased affinity for Arf-GDP on Golgi membranes (step 2). These reactions ultimately yield the formation of tubules containing GALNTs (step 3) and the enzymes' relocation to the endoplasmic reticulum (ER) where they glycosylate resident and neo-synthesized substrates (step 4). (**B**) Detailed representation of GBF1 phosphorylation: we hypothesize an interaction between Sec7 and HDS1, which is released by Y898 phosphorylation. Y876 phosphorylation affects the fold of the Sec7 domain itself, apparently inducing a partial melting and favoring binding to the Arf protein. This enhanced interaction leads to increased production of Arf-GTP.

activity of the kinase, transducing Src activation into changes in cell surface protein glycosylation. Specific inhibitors of Src/GBF1 interaction might thus have a therapeutic interest.

While phospho-GBF1 is not very abundant, Src can phosphorylate multiple sites on the GEF. GBF1 is a large protein with multiple domains, which can interact with one another. For instance, the N-terminal Dimerization and Cyclophilin Domain (DBS) interacts with the Homology Upstream of Sec7 domain (HUS) (*Ramaen et al., 2007*). The phosphorylation of the tyrosines in positions 317, 377, and 515 could alter this intramolecular interaction; while the five tyrosines in the Homology Downstream of Sec 7 domain 2 and 3 (HDS2/3) could affect other intra- or intermolecular interactions.

Mass spectrometry data and residue conservation analysis suggest that Y876 and Y898, located at and near the C-terminus of Sec7 domain, are particularly important. Based on the modeling efforts, Y876 phosphorylation induces the partial melting of an alpha-helix within the Sec7d, allowing for better binding to Arf. Partial melting is dependent on the phospho group interacting with either residues R843 or K844. Consistently, an R843A.K844A double mutant does not increase binding to Arf1 nor Arf1-GTP levels following Src activation. The R843E.K844E mutant is also unable to respond to Src, and in addition, it induces a reduced basal Arf-GTP level. This is consistent with the Sec7d of this mutant being locked into a low Arf-binding conformation. Y876 is conserved in all the Sec7d proteins we looked at. For several other GEFs, phosphorylation of the corresponding residue has been reported in various high-throughput databases. These findings suggest that the helix melting regulation is shared by other GEF proteins and could be a widespread mechanism for tyrosine kinases to regulate the GEF family of proteins.

The effect of Y898 phosphorylation is more hypothetical; it might release an inhibitory interaction between the Sec7d and the HDS1 domain (*Figure 7B*). However, to our knowledge, such interaction has not been reported. Notwithstanding, the data suggest that phosphorylation on both residues synergize to stimulate binding of GBF1 to Arf1-GDP. Y898 residue is highly conserved among vertebrates and invertebrates homologues of GBF1, suggesting that its phosphorylation is an ancient, conserved mechanism of regulation.

The importance of Y876 and Y898 phosphorylation is confirmed by the induction of tubules by the double phosphomimetic mutant. The tubules could only be observed at early time points after transfection of the mutant expression. At later time points, we observed a fragmentation of the Golgi and reduction of COPI staining. Similarly, we observed a peak of tubules formation shortly after Src activation (20–30 min). This kinetic control of Src allows distinguishing its short- to long-term effects on the Golgi. Like the phosphomimetic GBF1, Src activation tends to fragment the Golgi and reduce COPI staining.

The tubules are individually transient but extended, often measuring several micrometers long. Tubules are enriched in GALNT but depleted in the Golgi enzyme B4GAL-T1 and Giantin. By contrast, we did not observe an increase in COPI-enriched structures or vesicles in the Golgi vicinity after Src activation. The data thus suggest that GALNTs relocation is mediated by tubular transport carriers. Several groups have reported tubules-derived transport carriers emanating from the Golgi (*Bottanelli et al., 2017*; *Sengupta et al., 2015*). In a recent study, live super-resolution microscopy revealed COPI cups on Golgi membranes with surprisingly no evidence of COPI vesicles detaching (*Bottanelli et al., 2017*). Perhaps these results are to be compared with the proposal that ER to Golgi traffic is mediated by tubular transport carriers rather than COPII-coated vesicles (*Raote and Malhotra, 2019*; *Shomron et al., 2021*; *Weigel et al., 2021*).

How GBF1 phosphorylation might induce these tubules remains to be established. Our results indicate an increase in GBF1 affinity for Arf1-GDP, and mutagenesis of GBF1 suggests that this affinity increase is critical for the relocation (e.g., mutant R843A,K844A effect on Tn increase). GBF1 is recruited to Golgi membranes at the time of tubule formation. As shown previously, GBF1 Golgi membrane binding recruitment is dependent on Arf-GDP (*Quilty et al., 2018*). So, a possible model is that a membrane-tethered GBF1-Arf-GDP complex could recruit motors to pull on the membrane. Interestingly, the GBF1/Arf1 complex has been shown to interact with the microtubule motor Miro at mitochondria (*Walch et al., 2018*).

GBF1 is involved in different physiological situations in addition to the regulation of GALNTs activity (*Kaczmarek et al., 2017*). The axis Src-GBF1-Arf might also be involved in the regulation of Golgi organization based on the work by Luini's group (*Consoli et al., 2012*; *Luini and Parashuraman, 2016*; *Pulvirenti et al., 2008*). GBF1-Arf are also involved in the positioning of mitochondria, a locale

where Src kinase has also been detected (*Ackema et al., 2014*; *Hebert-Chatelain, 2013*; *Walch et al., 2018*).

We observed a wave of Arf-GTP production after Src activation. This burst of Arf-GTP could, alternatively, drive GALNT-containing tubule formation as it was previously shown that Arf-GTP can deform membranes (*Krauss et al., 2008*). We interpret the increase in Arf-GTP as phosphorylation facilitating the formation of a GBF1-Arf-GDP complex, while, based on the simulation, phosphorylation may not affect the GTP exchange catalytic reaction. The nucleotide exchange reaction can indeed be described with a $K_D$ for the binding of both proteins and a $k_{cat}$ for the conversion. Phosphorylation might regulate the $K_D$ without affecting the $k_{cat.}$ To note, overexpression of GBF1 alone increases Arf-GTP levels, indicating that GBF1 is a limiting factor and supporting the idea that an increase in affinity would result in higher Arf-GTP level.

Similarly to tubule formation, Arf1 and GBF1 recruitment on membranes and Arf1-GTP levels peak within 10–30 min after Src activation, suggesting a self-limiting mechanism. GBF1 phosphorylation itself is sustained over hours. Membrane-bound Arf-GDP, which is important for GBF1 recruitment, is probably also not the limiting factor because total membrane-bound Arf increases significantly after Src activation. The limiting factor may arise from the export of GALNTs-containing carriers, resulting in the depletion of an unidentified receptor on Golgi membranes. Indeed, Arf-GDP requires a protein receptor to bind to Golgi membranes (*Donaldson and Jackson, 2011*). Two candidates have even been proposed, the p23 protein and the SNARE membrin (*Gommel et al., 2001*; *Honda et al., 2005*). GBF1 also appears to require an additional receptor to bind efficiently to Golgi membranes (*Quilty et al., 2018*). It is possible that after Src activation either of these receptors (or both) are transported from Golgi membranes by the tubules-derived carriers. This could explain the peak in Arf-GTP levels and why overnight Src expression, while inducing a marked GALNT relocation, also results in a reduction of Arf-GTP levels: the Arf receptor may have been markedly depleted at the Golgi. By contrast, wild-type GBF1 overexpression, not promoting GALNTs relocation, would not induce this depletion and can thus stably increase Arf-GTP levels.

To sum up, our study reveals a key mechanistic insight into how Src regulates Golgi to ER retrograde traffic. It suggests that GBF1 binding to Arf-GDP plays a driving role in the formation of transport carriers. The understanding of this complex transport mechanism might help interfere therapeutically with a process driving the invasiveness of solid tumor cancer cells.

## Materials and methods

### Cloning and cell culture

Wild-type HeLa cells were from V. Malhotra (CRG, Barcelona). HEK293T cells were a gift from W. Hong (IMCB, Singapore). NIH3T3 and NIH3T3-vsrc mouse fibroblast were a gift from X. Cao (IMCB). Cell lines were previously purchased from ATCC and were authenticated by the vendor and cell morphology. All cell lines were verified to be free of mycoplasma contamination. Cells were maintained in DMEM with 10% fetal bovine serum (FBS) except for HEK293T, which was cultivated in 15% FBS. All cells were grown at 37°C in a 10% $CO_2$ incubator. Plasmids encoding full-length wild-type chicken SRC and an E378G mutant were a gift from Roland Baron (Harvard Medical School, Boston, MA). Src8A7F construct was obtained from Philip Cole's laboratory (Johns Hopkins University School of Medicine). The human GALT-GFP construct corresponds to the first 81 AA of human GALT fused in frame with *Aequorea coerulescens* green fluorescent protein, allowing targeting of the chimeric protein to medial and trans cisternae. The construct was purchased from Clontech Laboratories, Inc, Human GALNT2 (NM_004481) was cloned from a cDNA library generated from HT29 cells. All constructs were cloned into entry vector pDONR221 (Invitrogen, Life Technologies Corporation, Carlsbad, CA), and subsequently, gateway destination vectors expressing either emGFP or mCherry tag as described in *Gill et al., 2010*. All constructs were verified by sequencing and restriction enzyme digests before use. HeLa and HEK293T cell lines stably expressing Src8A7F-CmCherry or GALNT2-GFP were generated by lentiviral infection as described in *Gill et al., 2013*, and subsequently, FACS sorted to enrich for mCherry- or GFP-expressing cells.

### Antibodies and reagents

HPL conjugated with 647 nm fluorophore, Alexa Fluor secondary antibodies, and Hoechst 33342 were purchased from Invitrogen. Anti-GALNT1 for immunofluorescence staining was a gift from U.

Mendel and H. Clausen (University of Copenhagen, Denmark). Anti-GBF1 antibody for IP was from BD Biosciences (Franklin Lakes, NJ). Anti-GBF1 (C-terminus), anti-Giantin, and anti-Arf1 were from Abcam (Cambridge, MA). Human recombinant growth factors PDGF and EGF were purchased from BD Biosciences. Imidazole was purchased from Sigma-Aldrich (St. Louis, MO). GGA3 PBD agarose beads were purchased from Cell Biolabs, Inc (San Diego, CA). GTP-trap agarose beads were purchased from ChromoTek GmbH, Germany.

## Automated image acquisition and quantification

The staining procedures were performed as described in *Chia et al., 2014*. Briefly, images were acquired sequentially with a ×20 objective on a laser scanning confocal high-throughput microscope (Opera Phenix, PerkinElmer Inc). Image analysis was performed using the Columbus Software (version 2.8.0). GFP and mCherry-expressing cells were selected based on the intensity cutoff of the top 10% of expressing cells. The HPL staining intensity of the selected cell population was quantified by drawing a ring region outside the nucleus that covers most of the cell area. The HPL intensity per cell of each well was quantified. Statistical significance was measured using a paired *t*-test assuming a two-tailed Gaussian distribution.

## High-resolution fluorescence microscopy

The procedures were performed as described in *Chia et al., 2014*. Briefly, cells were seeded onto glass coverslips in 24-well dishes (Nunc, Denmark) before various treatments. They were fixed with 4% paraformaldehyde-4% sucrose in D-PBS, permeabilized with 0.2% Triton-X for 10 min, and stained with the appropriate markers. This was followed by secondary antibody staining for 20 min before mounting onto glass slides using FluorSave (Merck). The cells were imaged at room temperature using an inverted confocal microscope (IX81; Olympus Optical Co. Ltd, Tokyo, Japan) coupled with a CCD camera (model FVII) either with a ×60 objective (U Plan Super Apochromatic [UPLSAPO]; NA 1.35) or ×100 objective (UPLSAPO; NA 1.40) under Immersol oil. Images were processed using Olympus FV10-ASW software.

## High-resolution live imaging

For imaging of GALNT tubules, cells were seeded on eight-chamber glass chambers (Thermo Scientific, #155411) and acquired on LSM800 Zeiss inverted microscope with 37°C environmental chamber and using a ×63 objective under Immersol oil. Images were acquired at 4 s per frame for at least 30 min. For imidazole treatment of HeLa-IS cells expressing GALNT2-GFP, an equal volume of 2× concentrated imidazole in cell culture media was added dropwise to obtain a final concentration of 5 mM imidazole. For the experiment involving GBF1-depleted cells, cells were siRNA knockdown for 48 hr prior imaging. For the experiment involving phosphomimetic YE mutants, cells were transiently transfected with GBF1 plasmids and allowed to be expressed for short durations of 4–16 hr before imaging.

For the imaging of GALNT tubules under PDGF stimulation, cells were seeded in six-channel μ-Slide slides (ibidi GmbH, Germany) and treated with 50 ng/ml of PDGF stimulation using a perfusion pump system (ibidi GmbH) to inject the media at a constant and gentle flow rate. The cells were placed in a 37°C environmental chamber and imaged using an inverted confocal microscope (IX81; Olympus Optical Co. Ltd) coupled with a CCD camera (model FVII) with a ×100 objective (UPLSAPO; NA 1.40) under Immersol oil. Images were processed using Olympus FV10-ASW software.

## Immunoprecipitation and western blot analysis

Procedures for cell harvesting and processing for IP and western blot were performed as described previously with some modifications (*Gill et al., 2010*). For imidazole treatment and growth factor stimulations, cells were serum-starved for 24 hr before treatment with 5 mM imidazole, 50 ng/ml PDGF, or 100 ng/ml EGF, respectively. Cells were washed twice using ice-cold D-PBS before scraping in ice-cold RIPA lysis buffer (50 mM Tris [pH 7.4], 150 mM NaCl, 1% NP-40 alternative, cOmplete Protease Inhibitor, and phosphatase inhibitor [Roche Applied Science, Mannheim, Germany]). The lysate was incubated on ice for 30 min with gradual agitation before clarification of samples by centrifugation at 10,000 × *g* for 10 min at 4°C. Clarified lysate protein concentrations were determined using Bradford reagent (Bio-Rad Laboratories, Hercules, CA) before sample normalization. To IP endogenous GBF1,

samples were incubated with 2.5 µg of GBF1 (BD Biosciences) for 1 hr at 4°C with constant mixing. The IP samples were then incubated with 20 µl of washed protein A/G–Sepharose beads (Millipore) for 2 hr at 4°C with constant mixing. IP samples were washed three times with 1 ml of RIPA buffer containing cOmplete Protease Inhibitor and phosphatase inhibitor. For NGFP-GBF1 IP, clarified cell lysates were incubated with GTP-trap agarose beads (ChromoTek GmbH) for 2 hr before washing with GFP wash buffer (10 mM Tris [pH 7.5], 150 mM NaCl, 0.5 mM EDTA, cOmplete Protease Inhibitor, and phosphatase inhibitor) for three times. For Arf1-GTP loading assay, the clarified cell lysates were incubated with GGA3 PBD agarose beads (Cell Biolabs Inc) for 1 hr at 4°C with agitation before washing. Samples were diluted in lysis buffer with 4× SDS loading buffer and boiled at 95°C for 10 min. The proteins were resolved by SDS-PAGE electrophoresis using Bis-Tris NuPAGE gels (Invitrogen) and transferred to PVDF or nitrocellulose membranes. Membranes were then blocked using 3% BSA dissolved in Tris-buffered saline with tween (TBST: 50 mM Tris [pH 8.0, 4°C], 150 mM NaCl, and 0.1% Tween 20) for 1 hr at room temperature before incubation with primary antibodies overnight. Membranes were washed three times with TBST before incubation with secondary HRP-conjugated antibodies (GE Healthcare). Membranes were further washed three times with TBST before ECL exposure.

## In vitro Arf1 binding assay

Procedures for cell harvesting and processing for IP and western blot were described as above. NGFP-GBF1 expressed in cells was IP with GTP-trap agarose beads and washed three times with GFP wash buffer in the presence of cOomplete Protease Inhibitor and once with HKMT buffer (20 mM HEPES, pH 7.4, 0.1 M KCl, 1 mM MgCl$_2$, 0.5% Triton X-100) containing cOmplete Protease Inhibitor and phosphatase inhibitor. The purified NGFP-GBF1 on agarose beads were then incubated with either 4 mg of cell lysates of HEK293T cells expressing Arf1-V5 (agarose beads pre-cleared) or 10 µg recombinant Arf1-del17 protein for 1 hr for 4°C. Subsequently, the beads were washed three times with the HKMT buffer to remove unbound Arf1 protein. The beads were boiled at 95°C for 10 min. The amount of GBF1 bound Arf1 was resolved by western blotting.

## LC/MS analysis

The GFP-GBF1 bands of immunoprecipitated samples run on an SDS-PAGE using a NuPAGE 4–12% Bis-Tris gel (Invitrogen) were excised followed by in-gel digestion as described previously (*Shevchenko et al., 2006*). The peptide samples were subjected to a LTQ Orbitrap classic for data-dependent acquisition and a Q-Exactive for parallel reaction monitoring (PRM) (Thermo Fisher Scientific) analysis as described previously (*Swa et al., 2012*).

For PRM on Q-Exactive, targeted MS2 was carried out using a resolution of 17,500, target AGC values of 2E5 with maximum injection time of 250 ms, isolation windows of 2 Th, and a normalized collision energy of 27. MS/MS scans started from *m/z* 100.

## Data processing and database search

Raw file obtained from data-dependent acquisition was processed using Mascot Daemon (version 2.3.2, Matrix Science). Data import filter for precursor masses ranged from 700 to 4000 Da, with a minimum scan per group of 1 and a minimum peak count of 10. Mascot search was performed using the IPI Human database (ipi.HUMAN.v3.68.decoy.fasta or ipi.HUMAN.v3.68.decoy.fasta), trypsin as enzyme, and two allowed missed cleavages. Carbamidomethyl (C) was set as a static modification while the dynamic modifications were acetyl (Protein N-term), oxidation (M), and phosphorylation (S/T/Y). Tolerance for the precursor masses was 7 ppm and for fragments 0.5 Da for samples analyzed on LTQ Orbitrap.

Raw file obtained from PRM was processed using open-source Skyline software tool (Maclean, B. et al. Bioinformatics 2010, 26, 966; http://skylinemaccosslab.org.). The accuracy of the peaks assigned by Skyline was manually validated using Thermo Xcalibur Qual Browser by manual inspection of the targeted MS2 spectra and by XIC to ensure the *m/z* of the fragment ions are within 20 ppm of their theoretical values.

## Structure modeling and molecular dynamics

The 3D structure of GBF1 protein is not available; therefore, a structural model of the Sec7 domain of GBF1 (GBF1_Sec7) protein was generated using comparative modeling methods (*Sali and Blundell,*

*1993*). Homology model of the GBF1_Sec7 in its autoinhibited form was generated using the crystal structure of the autoinhibited form of Grp1 Arf GTPase exchange factor (PDB: 2R0D, resolution 2.0 Å), which shares ~65% homology with GBF1 in the Sec7 domain. A 3D structural model of the GBF1_Sec7-Arf1 complex was generated using the crystal structure of Arno_Sec7-Arf1 (PDB: 1R8Q, resolution 1.9 Å) since Arno shares ~65% homology with GBF1 in the Sec7 domain.

MD simulations were carried out with the pemed.CUDA module of the program Amber18 (*Case et al., 2018*) using standard and well-tested protocols (*Kannan et al., 2015*). All atom versions of the Amber 14SB force field (ff14SB) (*Maier et al., 2015*) were used to represent the protein. Force field parameters for phosphorylated tyrosine and GTP were taken as described elsewhere (*Homeyer et al., 2006*); an overall charge of –2e is assigned to the phosphate groups. The Xleap module was used to prepare the system for the MD simulations. All the simulation systems were neutralized with appropriate numbers of counterions. Each neutralized system was solvated in an octahedral box with TIP3P (*Jorgensen et al., 1983*) water molecules, leaving at least 10 Å between the solute atoms and the borders of the box. All MD simulations were carried out in explicit solvent at 300 K. During the simulations, the long-range electrostatic interactions were treated with the particle mesh Ewald (*Darden et al., 1993*) method using a real space cutoff distance of 9 Å. The SETTLE (*Miyamoto and Kollman, 1992*) algorithm was used to constrain bond vibrations involving hydrogen atoms, which allowed a time step of 2 fs during the simulations. Solvent molecules and counterions were initially relaxed using energy minimization with restraints on the protein and inhibitor atoms. This was followed by unrestrained energy minimization to remove any steric clashes. Subsequently, the system was gradually heated from 0 to 300 K using MD simulations with positional restraints (force constant: 50 kcal mol$^{-1}$ Å$^{-2}$) on the protein atoms over a period of 0.25 ns allowing water molecules and ions to move freely. During an additional 0.25 ns, the positional restraints were gradually reduced followed by a 2 ns unrestrained MD simulation to equilibrate all the atoms. Production runs were carried out for 250 ns in triplicates (assigning different distributions of initial velocities) for each system. Simulation trajectories were visualized using VMD (*Humphrey et al., 1996*), and figures were generated using PyMOL.

Binding free energies and per-residue decomposition of binding free energies between the GBF1_Sec7 (unphosphorylated and phosphorylated at Tyr876) and Arf1 were calculated using the standard MMPBSA approach (*Homeyer and Gohlke, 2012*; *Hou et al., 2011*). Conformations extracted from the last 125 ns of the MD simulations of each GBF1_Sec7-Arf1 complex were used, and binding energy calculations/per residue decomposition analysis were carried out using standard protocols (*Kannan et al., 2015*). BSA was computed using the program NACCESS (*Hubbard and Thornton, 1993*).

# Additional information

### Competing interests

Frederic A Bard: Reviewing editor, *eLife*. The other authors declare that no competing interests exist.

### Funding

| Funder | Grant reference number | Author |
| --- | --- | --- |
| Astar | Core fund | Frederic A Bard |

The funders had no role in study design, data collection and interpretation, or the decision to submit the work for publication.

### Author contributions

Joanne Chia, Conceptualization, Data curation, Investigation, Methodology, Project administration, Validation, Visualization, Writing – original draft, Writing - review and editing; Shyi-Chyi Wang, Data curation, Investigation, Methodology, Writing – original draft; Sheena Wee, Data curation, Investigation, Methodology; David James Gill, Conceptualization, Investigation; Felicia Tay, Investigation, Methodology; Srinivasaraghavan Kannan, Investigation, Methodology, Visualization; Chandra S Verma, Conceptualization; Jayantha Gunaratne, Conceptualization, Methodology, Supervision; Frederic A

Bard, Conceptualization, Funding acquisition, Supervision, Writing – original draft, Writing - review and editing

## Author ORCIDs

Joanne Chia http://orcid.org/0000-0002-6617-0278
Jayantha Gunaratne http://orcid.org/0000-0002-5377-6537
Frederic A Bard http://orcid.org/0000-0002-3783-4805

## Decision letter and Author response

Decision letter https://doi.org/10.7554/eLife.68678.sa1
Author response https://doi.org/10.7554/eLife.68678.sa2

## Additional files

### Supplementary files

• Transparent reporting form

• Source data 1. Original full raw unedited gels or blots in all figures and figure supplements. The red boxes indicate the regions of the blot presented in the figures of the article.

• Source data 2. Raw quantification data and statistical tests performed in all figures and figure supplements.

### Data availability

Source data of western blots and all quantifications have been provided for all figures.

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
