## [Editor Report]

The Src tyrosine kinase controls cancer-critical protein glycosylation through Golgi to ER relocation of a subset of Golgi enzymes, GALNTs, from the Golgi to the ER. The authors show here that Src induces the formation of tubular transport carriers containing GALNTs by phosphorylating GBF1 and promoting its binding to Arf1. This study presents some of the first clues to the molecular events underlying Src-regulated relocalization of glycosyltransferases.

---

## [Decision Letter]

**Decision letter after peer review:**

[Editors’ note: the authors submitted for reconsideration following the decision after peer review. What follows is the decision letter after the first round of review.]

Thank you for submitting your work entitled "Src activates retrograde membrane traffic through phosphorylation of GBF1" for consideration by *eLife*. Your article has been reviewed by 3 peer reviewers, and the evaluation has been overseen by a Reviewing Editor and a Senior Editor (Suzanne Pfeffer). The reviewers have opted to remain anonymous.

Our decision has been reached after consultation between the reviewers. Based on these discussions and the individual reviews below, we regret to inform you that your work will not be considered further for publication in *eLife* at this stage.

Overall, the reviewers were excited by the hypothesis that Src activation and phosphorylation of GBF1 control specialized retrograde Golgi-ER trafficking. That being said, they raised a number of issues that would take some time to address, to support fully the conclusions of the present study. One of the reviewers provided the following comments that would be important to consider in regard to your model.

"It seems that the authors are claiming that the phospho-GBF1-Arf1-GDP complex is more stable (which their data do suggest). But their data do not clearly discriminate between positive or negative regulation…. If Src is a positive regulator of GBF1, then why does phospho-GBF1 form stable complexes with Arf1-GDP? A GEF that binds *more* stably to Arf1-GDP is actually *less* active. This may be counterintuitive, but this is how GEFs work: the action of BFA and the catalytic E->K mutations in Sec7 domains are both good examples of how when the GEF domain binds more strongly to Arf1-GDP, they become worse GEFs…If they were trying to demonstrate that Src is a positive regulator of GBF1, the data are not convincing. They have shown that *cellular* levels of Arf1-GTP transiently increase, but they have not shown that Golgi-localized Arf1-GTP increases, and there are other ArfGEFs in cells. The reduction of COPI localization provides further evidence that Arf1-GTP is not increasing at the Golgi.

The key concept is that if a GEF bound strongly to GTPase-GDP, it would actually inhibit activation by GTP binding. Here is a reference from Antonny and Cherfils (and Chabre) for the specific example of a Sec7 domain mutant that binds to Arf-GDP better and therefore is a worse GEF: https://pubmed.ncbi.nlm.nih.gov/9649435/

Here is a review from Wittenhofer: https://pubmed.ncbi.nlm.nih.gov/17540168/

in which he key part about affinities is: "the affinities of the binary complexes between the G protein and either the nucleotide or its GEF are very high. In contrast, the affinities of the exchange factor for the nucleotide-bound G protein and of the nucleotide for the exchange-factor-bound G protein (the ternary complexes) are much lower.

This is consistent with the energetics of how all enzymes work – enzymes bind best not to their substrates or products, but to their transition states (otherwise enzymes would simply stabilize their substrates rather than performing catalysis). The nucleotide-free state of GTPases is the intermediate that is closest to the transition state of the exchange reaction. There is a general misconception that GEFs bind best to GDP-bound GTPases, but this is not true and has arisen because the so-called "GDP-locked" mutants that people have used actually have a reduced affinity for nucleotides (both GDP and GTP) and therefore tend towards being nucleotide free. Anyway, it's a common misconception and usually one that is not too important but in this case it matters for their mechanism."

Another reviewer wrote, "Since simply seeing more Arf1 at the Golgi does not necessarily mean that it is active, perhaps they could perform a FRAP experiment? Given that the Golgi is a large organelle, the kinetics of recovery would reflect association of ARF1 to the Golgi. If Src really increases the levels of active Arf1 at the Golgi, then it should affect the on/off-kinetics in a FRAP experiment. Or they could do Trp-fluorescence assays with recombinant Arf1 and GBF1."*Reviewer #1:*

In this study from the Bard lab, the authors explore the mechanism behind Src activation induced relocalization of GALNTs enzymes from Golgi to ER. They use an imidazole-inducible Src model, which generates increased O-GALNacylation. In response to Src activation, they observe GFP-GALNTs in tubules emanating from the Golgi. They also observe more Arf1-GTP in Src-activated cells. They Identify Y-phosphorylation sites in GBF1 induced by Src phosphorylation. They can confirm Src-induced phosphorylation at one of these sites (Y876) by phospho-specific antibody. They find that expression of the Y876F GBF1 mutant blocked the Src-activation induced increase in cellular Arf1-GTP levels. Expression of the Y876E/Y898E mutant dramatically reduced the amount of basal Arf1-GTP. They perform modeling studies suggesting that phosphorylation of Y876 should increase the affinity of the GBF1 Sec7d for Arf1.

Their observations of Src activation are similar to the acute effects of BFA. BFA triggers formation of stable, complexes between GBF1 and Arf1-GDP. These complexes bind stably to the Golgi, but are inactive and effectively poison the GEF, reducing the amount of Arf1-GTP generated at that site. Indeed, Src activation results in slight depletion of COPI at the Golgi, and increase in GBF1 at the Golgi, similar to effects of BFA. BFA also triggers tubules emanating from the Golgi.

The authors conclude with a model in which Src phosphorylation of GBF1 results in stable GBF1-Arf1 complexes that somehow generate tubules from the Golgi that traffic GALNTs enzymes back to the ER.

I do not find the overall mechanistic explanation of the observations to be convincing, and in my mind the authors have not ruled out alternative explanations. While the authors have made some interesting observations, I do not believe they have managed to correctly connect the results to a coherent and plausible mechanism.

The authors' model does not even appear to be internally consistent – on the one hand they are claiming that Src activation results in GBF1 activation of Arf1 at the Golgi. On the other hand, they are claiming that Src activation results in stable GBF1-Arf1-GDP complexes that would not be competent for activating Arf1.

In summary, the authors can't seem to decide on whether Src is a positive or negative regulator of GBF1, and it remains unclear whether the resulting GBF1-Arf1 complexes are actually responsible for trafficking GALNTs enzymes from the Golgi to the ER.

Specific issues with interpretations and experimental design:

The authors find that overexpression of GBF1 increases the amount of Arf1 activated in cells in response to Src activation, but critically, they do not test the dependence of Src-responsive Arf1 activation on GBF1. Why have the authors not monitored the extent of Src-triggered Arf1 activation in GBF1-knockdown cells?

The authors claim that Src activation increases the amount of Arf1 at the Golgi but the fluorescence images do not appear to support this claim. Although Arf1 is observed on the resulting tubules, this could represent its presence in non-productive membrane-bound complexes with GBF1.

The authors draw the unsupported conclusion that "these GBF1-Arf1-GDP complexes are directly involved in BFA-induced tubule formation." It would be straightforward for the authors to test whether GBF1 is indeed required for formation of the Src-induced tubules. Similarly, it would be even more convincing if they could also determine whether GBF1 is required for BFA-induced tubules (if this has not already been demonstrated in the literature).

Furthermore, the authors need to test the effects of GBF1 Y/F and Y/E mutants on Src-induced tubule formation.

Perhaps the observed increase in cellular Arf1-GTP levels could be due to the action of the BFA-independent GEFs, perhaps as a compensatory response to the loss of GBF1 activity induced by Src-dependent phosphorylation?

The authors make the claim "Altogether, these results indicate that phosphorylation on Tyrosines Y876 and Y898 drives an increase of affinity of GBF1 for Arf1-GDP, in turn increasing Arf1-GTP levels and promoting GALNTs relocation." Yet the phosphomimetic mutants have reduced Arf1-GTP levels! It appears more likely that phosphorylation is inducing a BFA-like effect, and not actually increasing GBF1 activation of Arf1.

The authors results, both experimental and modeling, strongly suggest that phosphorylation results in stable, non-productive GBF1-Arf1 complexes that would not lead to an increase in Arf1 activation. For an exchange factor, a mutation or PTM that increases affinity for the GTPase substrate will actually result in reduced activation as the GEF needs to dissociate from the GTPase in order for GTP to stably bind.

A key question that is not addressed is whether Src phosphorylation of GBF1 triggers increased Golgi to ER transport or decreased ER to Golgi transport of GALNTs? Either of these situations could cause the observed response to Src activation assuming the GALNTs normally cycle between the Golgi and ER.*Reviewer #2:*

In this manuscript, Chia et al. tried to provide a first glimpse of the molecular machinery driving the GALNTs Activation (GALA) pathway, which was proposed by the Bard group ten years ago. The major claim of the paper is that the regulated Src-dependent phosphorylation of GBF1 (specifically on Y876 and Y898) is the primary molecular switch that drives formation Golgi-originated membrane tubules that serve to deliver GALNT enzymes to ER. The experiments performed in the manuscript are appreciable and abundant. However, the presentation and writing are sloppy, which has made comprehension tiresome and also tricky. The discussions are shallow, and the authors have conveniently avoided rationalizing interesting/surprising findings. Overall, the hypothesis that Src activation and phosphorylation of GBF1 control specialized retrograde Golgi-ER trafficking looks exciting and valid, but this needs to be supported by more robust data, which is unfortunately not presented in this manuscript. A substantial revision would be required for this manuscript to be published.

1. A major assertion throughout this manuscript is the regulated relocalization of the enzymes GALNTs from Golgi to ER, and corresponding HPL staining has been used as a readout of GANLT activity. Unfortunately, all the images of HPL staining throughout the paper are of low quality and have inconsistencies in the staining pattern and intensity. It is essential to provide high-resolution images and quantify colocalization of HPL signal with ER and Golgi markers. Colocalization of the endogenous GALNT1 with these markers will be a more accurate measure of the relocalization of the enzyme to the ER. PDI IP with VVL was a smart approach and indeed suggest GALNT activity in the ER but could also indicate minor relocalization of PDI4A to the Golgi compartment. The effect of the imidazole treatment on intracellular localization of PDI should be tested.

In F1b, colocalization of GALNT with an ER marker will substantiate this finding. This figure also shows a noticeable increase in the perinuclear (Golgi?) staining intensity of HPL, which seems to have been ignored. Does the relocalization of the enzyme to the ER have anything to do with its increased activity in the Golgi?

2. Another major inconsistency in the manuscript is related to the imidazole-inducible activity of Src8A7F. This activity is depicted as pY 4G10 blots, but these blots (Figures2E, 3A, 4A, S1B) are all different and inconclusive. Random blot parts ( 40-180 kDa, 70-180 kDa, 60-120 kDa) are shown. It is imperative to quantify Src activity. Hopefully, careful Src quantification will shed some light on the unexplainable temporal activation of Arf1 protein.

3. The third major problem is related to the use of overexpressed tagged proteins utilized thought the study. This approach was more or less valid ten years ago, but now with advances of CRISPR and other gene-editing tools, it is possible to avoid (or at least control for) artifacts connected with protein overexpression and tagging. Authors show that overexpressed GALNT2-GFP and Arf1-V5 entering into the tubular compartment emerging from the Golgi (F2C), but they failed to show a similar pattern for the endogenous GALNT1 (S1L). Moreover, the activation pattern of the endogenous Arf1 (F2E) is very different from the activation dynamic of Arf1-V5 (F3A). All these inconsistencies should be adequately addressed in the text.

4. In S1F, how was the experiment performed? What is "control 0h" and what are the other time points grouped as "24h Imdz wash"? This figure needs to be properly labeled.

5. The localization of Src in S1D is different than that of Src8A7F. Does this mutation affect its localization?

6. S1H has cells missing ManII. Is there any effect of Src activation on ManII?

7. In F1D is a smart approach to measure GALNT activity in the ER. An increase in glycosylation of PDI with time upon Src activation, but in later, it is shown that Aft-GTP bursts occur within 5-10mins after Src activation and in SrcEG mutant, Arf is depleted. This data is at odds with the prolonged effect of Src activation on the HPL staining intensity and GALNT relocalization. Moreover, the burst of GBF1 phosphorylation is seen at 120 mins in F4AandB. How do all this fit in into the prolonged effect on GALNT activity and increased HPL staining?

8. Overexpressed GALNT2 indeed shows tubules, but the endogenous GALNT1 tubules are absent in S1L. In this figure, βCOP staining is also very odd. The localization of βCOP to the Golgi is well established, and there are quite a few suitable antibodies that work well (https://doi.org/10.1111/j.1600-0854.2008.00724.x). Such poorly quality staining cannot be used to rule out the involvement of COPI in GALNT relocalization.

9. For F2, which shows GALNT1+2, Arf1 and Afr3 KDs provide data on the efficiency of the KD. Also, look at the effect of Afr1 KD on the localization of active GBF1. Is Arf1-GDP required for the membrane recruitment of GBF1?

10. Arf-V5 and endogenous Afr do not follow the same trend. Please repeat GGA IP in F3A and probe for endogenous Arf.

11. Is GBF1 the only Arf GEF that is phosphorylated upon Src activation? Is it not likely that BIGI, ARNO and BRAG2's conserved tyrosine residues would be phosphorylated too once Src is activated and these activated GEFs would have some contribution to Golgi trafficking?

12. In F4E, Src has been claimed to directly phosphorylate GBF1. However, during the purification of GFP-GBF1 there could be other associated proteins that may be effectors of activated Src. This needs to be acknowledged and before claiming that Src kinase directly phosphorylated GBF1.

13. Another significant discrepancy in this study is the effect of the different Src mutants on the HPL staining and Src localization/profile itself. The model in figure 7 fails to explain how the short burst of pGBF1 and AFR1-GTP triggers prolonged relocalization of GALNT. The final model is unclear and somewhat misleading. It seems to indicate that tubule formation is caused by the tight Arf1-GBF1 complex, but the data show a transient burst of Arf1-GTP (not bound to GBF1) as a major cause of tubule formation. What selects GALNTs into these tubules*Reviewer #3:*

The manuscript by Chia et al. is one of a series of elegant manuscripts by the Bard lab on the GALA pathway. The topic is very interesting and I think it fits to the scope of the journal. The data are mostly of very high quality and the finding is novel, as it explains the role of Src in retrograde transport.

Apart from some technical and minor comments that I mention below, I am mainly concerned with one point: the authors claim that Src induces retrograde transport that is dependent on Arf1 and GBF1. However, the claim that it is COPI-independent. The evidence for this is relatively weak. Regardless of this, there is no evidence that the tubules that the tubules observed in response to Src activation are directed towards the ER. I did not see evidence that these tubules move towards the ER, or whether they passage through the ERGIC. Maybe these tubules detach from the Golgi to form a carrier that moves to the ER.

I am happy to drop this point, if the editor and/or the other reviewers think that this is beyond the scope of this paper.

1. The images in Figures 1B and S1C are identical. This should be clearly indicated. The authors should state that S1C shows the very same cells as 1B, only with a co-staining for Giantin.

2. I noticed that the size of the Golgi (and the cell) is bigger in imidazole-treated cells (with Src activation). Since the increase of fluorescence is mostly apparent in the Golgi region, I think that the HPL intensity should be normalized by the size of the cell.

3. Figure 1G: I think that the authors should image more than just 4 cells. It is also not indicated how many experiments were performed.

4. Figure 2C: this result requires some form of quantification. How many tubules were observed and in how many cells? How many of the tubules were Arf1-positive?

5. The conservation of tyrosines (in GBF1) from yeast to mammals is meaningless. Yeast (and fungi in general) have no tyrosine kinases (there are very few exotic exceptions, but *S. cerevisiae* is definitively negative). The fact that yeast has no tyrosine kinases actually should prompt to investigate the tyrosine residues that are not conserved. I think this passage should be re-written. As it is scientifically not accurate and misleading.

6. Figure 5: the authors state that the stimulus was "nearly abolished". This is not correct. Looking at the blot, I would. Re-word this and rather use "reduced". The double mutant is also still phosphorylated

7. Figure 5F is missing. Figure 5E just shows the quantification, but no primary data.

[Editors’ note: further revisions were suggested prior to acceptance, as described below.]

Thank you for resubmitting your work entitled "Src activates retrograde membrane traffic through phosphorylation of GBF1" for further consideration by *eLife*. Your revised article has been evaluated by Suzanne Pfeffer (Senior Editor) and a Reviewing Editor.

The manuscript has been improved but there are some remaining issues that need to be addressed, as outlined below; the reviewers think you have very interesting results but need to provide more details related to the mechanism of what is observed.

The reviewers all agreed that the most critical thing missing is direct demonstration of GBF1-Arf1-GDP tubulation activity. They also suggest that you should consider alternative explanations for the data. Specifically, they suggest you load Arf1 with GDP-betaS and add phosphorylated GBF1, then test if the complex drives formation of tubules or carriers out of Golgi membranes.; they note that others have purified GBF1 and it may be possible to immunopurify a FLAG-tagged version of the protein expressed in 293T cells.

Alternatively, if you can show that inducible expression of GBF1 phospho-mimetic mutants Y876E and Y898E can drive Golgi to ER localization of GALNTs in the absence of Src activation and show the cargo selectivity of such tubules, that would be sufficient to justify presentation in *eLife*. This alternative would require a complete reframing of the text to indicate that an unknown mechanism yet to be elucidated accomplishes this unexpected but fascinating molecular process. Such reframing would be important as the reviewers remain rather skeptical about GBF mediated membrane tubulation.

I include their detailed responses to guide your next steps. They acknowledge that this may require more than a short time to complete but still want to give you a chance to try. I realize that this is not the answer you were hoping for and we will be flexible about the time required to respond.*Reviewer #1:*

The revised version of the manuscript by Chia et al. is only a minimally altered version of the original submission. The main changes in the text are within the discussion and the authors are mainly to argue against the criticism raised by the reviewers.

I find the following points confusing:

1. The authors state that they don't think that it is a key point that Arf1-GTP is at the Golgi. They propose that it is the complex between GBF1 and Arf1-GDP that is relevant for tubule formation. I am not sure that the experimental evidence that is provided is convincingly showing this.

2. The authors argue that it is irrelevant to talk about COPI, because Arf1 could generate tubule itself. They cite a paper by Francesca Bottanelli. I would like to stress that the Botallei paper showed Arf1 tubules that are directed to the cell periphery and NOT retrograde transport to the ER. In addition, the Botanelli paper did not suggest that these are tubules generated with Arf1-GDP. Therefore, I find the argumentation used in the rebuttal a bit confusing.

3. I think that the point that the complex of Arf1-GDP and GBF1 generates tubules should be demonstrated experimentally.

4. The argument about the conservation of the tyrosine residue in yeast is confusing. Firstly, there is very little tyrosine phosphorylation in yeast. There are dual-specificity kinases in yeast that can perform tyrosine phosphorylation. However, there kinases are conserved in humans. So why do mammalian cells then use Src, and not the ancestral dual-specificity kinase. I find it confusing why the authors are insisting on keeping a piece of text that is so speculative and most likely wrong. Anyhow, this is just a minor point.

I think that the point that the complex of Arf1-GDP and GBF1 generates tubules should be demonstrated experimentally. Based on the new discussion and the rebuttal letter, I see that the authors themself consider this very important. I think this point deserves to be tested experimentally.

*Reviewer #2:*

Signaling pathways modify intracellular membrane trafficking and protein modifications on different levels. In this study, the authors investigate the mechanism behind Src activation-induced relocalization of a subset of Golgi enzymes, GALNTs, from Golgi to ER. In response to Src activation, authors observe GALNT2-GFP in Golgi-derived tubular structures. They also observed temporal activation of small GTPase Arf1 and phosphorylation of Arf1 GTP exchange factor, GBF1. The authors propose the model in which phosphorylation of GBF1 by Src results in GBF1-Arf1 complexes that generate membrane tubules for traffic GALNTs from Golgi to ER.

In the revised submission, Chia et al. have fixed the many errors in the manuscript, which has indeed improved its presentation. They also provided two new experiments and significantly updated the Discussion section. At the same time, authors mostly responded to our and other critiques not by performing requested experiments/controls but by referencing their own previously published work and by modifying the text. We believe that this kind of response is not adequate.

My main concerns are as follows:

1. As I have stated in the first round of review, to quantify Src-dependent Golgi-ER relocalization of Golgi enzymes, it is essential to provide reproducible, high-quality images quantify colocalization of HPL signal with ER and Golgi markers. The ratio of ER to Golgi signal is the most important parameter here. The work should be reproducible, and therefore mere references to previously published work are not sufficient. The model system (HeLa cells with inducible Src) is not adequately characterized in terms of relocalization of GALNTs from Golgi to ER. Specifically, images presented in Figure 1B and, even more strikingly, in Supplementary Figure 1C, H are not supporting the notion that ER/Golgi ratio of HPL signal (i.e. relocalization of GALNTs) has been changed significantly. Without proper verification that retrograde trafficking of Golgi enzymes is increased in a Src/GBF1-dependent manner, the title of the manuscript is not supported by the data since the direction of movement of GALNT-GFP-positive tubular structures is unknown.

2. Endogenous Golgi cargo has not been detected in Golgi-derived tubules, suggesting that tubule formation could be an artifact of protein overexpression. Authors' arguments that "tubules are transient in nature… it makes it harder to observe…chemical fixation significantly disrupts tubule integrity" are valid in general, but not at the level of *eLife* quality paper. Moreover, for the majority of tubule imaging in the manuscript (Figures 1G, 2C, S1K, S2A), the authors successfully used chemical fixation to demonstrate the association of overexpressed proteins with tubular structures. If necessary, consider live cell microscopy.

3. Authors clearly shown that Src phosphorylates GBF1, and they identified target phosphorylation sites on GBF1. Authors are suggesting that upon Src activation, GBF1 binds to Arf1-GDP, and this complex stimulates the formation of GALNT carrying tubules. However, I still have a hard time aligning this hypothesis with the data presented in the manuscript. During the burst of Arf-GTP, one would assume that k-on exceeds k-off resulting in Arf-GTP levels peaking and Afr-GTP exceeding GBF1-Arf-GDP. However, starting from 10 min following Src activation, the GALNT tubules emanating from the Golgi are significantly increased. This would indicate that tubule formation is not really driven by GBF1-Arf-GDP because it peaks at 20-30 min, when GBF1-Afr-GDP would be at its lowest. In S2B they show that Arf-GTP levels are lower than the controls. In S5AandB, Afr-GTP level as low as the control condition. Authors conclude that constitutively active SrcEG has the same effect on Arf-GTP levels as phosphor-mimetic GBF1 resulting in low levels of Afr-GTP. One can imagine a hypothetical scenario where the entire pool of GBF1 is engaged with Arf-GDP. But, SrcEG cells do have increased HPL staining in the ER and Golgi which, as the authors claim, is due to GALNTs transported to the ER in GBF1-Arf dependent tubules. This would indicate that the transport of GALNTs to the Golgi is independent of Arf-GTP, which is at odds with the kinetics of tubule formation and Arf-GTP levels in Figure 2.

4. As suggested by other reviewers, to validate the model that Src-dependent phosphorylation of GBF1 is causing relocalization of Golgi enzymes, it will be essential to show that inducible expression of GBF1 phospho-mimetic mutants Y876E and Y898E would drive Golgi to ER localization of GALNTs in the absence of Src activation.

*Reviewer #3:*

The authors model is now more clear, but still not convincing. They are proposing that GBF1-Arf1-GDP complexes are tubulating membranes. There is no precedent for such an activity and other plausible explanations have not been ruled out. As stated in the previous review, an alternative explanation is that their observations are similar to those observed under BFA treatment. The Hsu and Luini groups explored one possibility for why BFA induces Golgi tubulation: (https://pubmed.ncbi.nlm.nih.gov/21725317)

The authors model and cartoon for the GEF reaction in Figure 7A is too simplistic, as the step they label with "kcat" actually represents more than one step: first GDP must dissociate before GTP can dissociate. This is absolutely essential as GDP and GTP occupy the same binding site. Also, the use of "kcat" generally refers to the rate-limiting step, and this is exactly the point I am making – an increase in k_on_ (which appears to be the consequence of phosphorylation) is irrelevant to the overall reaction rate constant if k_on_ is not the rate-limiting step. The step labeled "kcat" could very well be rate-limiting (and at the very least there is no reason to conclude that it is not rate-limiting, which is what the authors appear to be claiming). Therefore, my original concern still stands: their data are most consistent with phospho-GBF1 forming a stable complex with Arf1-GDP, which will reduce, rather than enhance the kinetics of exchange.

I also note that in Figure 7, the authors have incorrectly used upper case 'K's, which are used for equilibrium constants, rather than lower case 'k's, which should be used for the rate constants that they are referring to. Furthermore, by convention kcat is used to refer to the overall rate constant of the reaction.

The authors claim in the rebuttal letter that the "kcat" step is unlikely to be rate-limiting because this is the case for "most enzymes in metabolic pathways acting on small molecules" is both unfounded and probably irrelevant to an exchange factor.

The authors claim that the Antonny, Chabre, and Cherfils paper supports their model but I strongly disagree. Yes, the mutant they used blocks exchange, and also stabilizes binding to Arf-GDP. The authors appear to be ignoring the fact that GDP must dissociate before GTP can bind. Strong binding to Arf-GDP will slow GDP dissociation, and therefore also slow the rate of exchange. The authors' strong language on these points does not make their logic any more correct.

The authors are twisting themselves in knots by explaining that their in vitro binding assay does not include GTP, rather than performing an actual exchange assay in which GTP is included. Rather than trying to argue with reviewers, they could simply perform an actual nucleotide exchange experiment to see whether phosphorylated GBF1 is a better GEF or not. Based on their proposed model for how tyrosine-phosphorylation within the Sec7-domain enhances GEF activity, this should be straightforward to perform using the Sec7-domain of GBF1, rather than the full-length protein which the authors note is difficult to purify.

Finally, from my perspective, I still don't understand why on the one hand, the authors are arguing that phosphorylation makes GBF1 a better GEF, yet on the other hand, the authors' model invokes a functional role for a stable GBF1-Arf1-GDP complex. Neither of these two possibilities is fully supported by the data.

[Editors’ note: further revisions were suggested prior to acceptance, as described below.]

Thank you for resubmitting your work entitled "Src activates retrograde membrane traffic through phosphorylation of GBF1" for further consideration by *eLife*. Your revised article has been evaluated by Suzanne Pfeffer (Senior Editor) and a Reviewing Editor.

The manuscript has been improved but there are some remaining issues that need to be addressed, as outlined here.

The reviewers felt that although SrcKM is supposed to be dominant negative, it is difficult to account for the effects of endogenous Src and other family members. To show that the phosphomimetic mutant of GBF1 is sufficient to induce tubule formation, the reviewers felt that the experiment should be carried out in Src-deficient cells, i.e. with pharmacological or genetic interference and not with a dominant negative approach. Reviewer 2 agreed that you would know best how to do that, as long as you can demonstrate that Src is not active under whatever treatment they use.*Reviewer #1:*

The authors have addressed some of the initial concerns I had. I also think that the inclusion of the new data with the phosphomimetic mutant of GBF1 strengthen the story as a whole. Furthermore, I think it was also good to remove the parts of the discussion on the conservation of the phosphotyrosine sites in yeast GFP1, because yeast has no tyrosine kinases. Finally, removing the part with the highly speculative role of Arf1-GDP in tubule formation was absolutely necessary. Although the current work leaves many open questions, I think it is an important contribution to the field and future work will focus on the outstanding questions.

*Reviewer #2:*

The previous review stated "If you can show that inducible expression of GBF1 phospho-mimetic mutants Y876E and Y898E can drive Golgi to ER localization of GALNTs in the absence of Src activation and show the cargo selectivity of such tubules, that would be sufficient to justify presentation in *eLife*. "

The authors have addressed concerns regarding the proposed tubulation mechanism by changing the text to be less specific about the details. The authors now provide evidence that expression of phospho-mimetic GBF1 induces Golgi tubules. However, they have not shown cargo selectivity of these tubules. They show these tubules stain with GALNT2, but do not show any selectivity.

Perhaps more importantly, unless I am missing it, they have not shown that phospho-mimetic GBF1 is sufficient to induce GALNT relocalization to the ER in the *absence* of Src activation. In fact, the effect of the phosphomimetic mutant appears more pronounced when Src is activated (Figure 5 supplement 1, panel F, blue bars) than when Src is not activated (Figure 5 supplement 1, panel F, red bars).

[Note that Figure 5 supplement 1 panels E and F labels suggest the phospho-dead mutant (Y->F) is being used but text and legend indicates phospho-mimetic (Y->E)]

This suggests that the relevant Src target is a protein other than GBF1.

Therefore, the functional importance of GBF1 phosphorylation by Src remains unconvincing.

---

## [Author Response]

[Editors’ note: the authors resubmitted a revised version of the paper for consideration. What follows is the authors’ response to the first round of review.]

Overall, the reviewers were excited by the hypothesis that Src activation and phosphorylation of GBF1 control specialized retrograde Golgi-ER trafficking. That being said, they raised a number of issues that would take some time to address, to support fully the conclusions of the present study. One of the reviewers provided the following comments that would be important to consider in regard to your model.

We thank all the reviewers for their evaluation of the manuscript and the detailed comments. We have taken into account their concerns and questions. We provide a conceptual rebuttal to the main concern of GBF1 activation versus Arf1 binding. In addition, we have performed the following additional experiments:

– GBF1 knock-down in Src activation conditions to show that it is indeed GBF1 that drives Arf1-GTP formation (Figure 3C-D).

– GBF1 knock-down and live imaging showing that tubules are not formed in the absence of GBF1: Figure S3G-H.

We have significantly re-written the Discussion to better explain our reasoning and its limits. We also added a diagram (Figure 7A) to explain our model of increased affinity with increased Arf-GTP production. We understand some aspects of our model may be counterintuitive or controversial, and we would like to point out the results seemed counterintuitive to us as well at first. This is one of the reasons why the paper has been in the works for several years. We hope the model will be more clear with this new version and that our study will finally find its home and an audience at *eLife*.

"It seems that the authors are claiming that the phospho-GBF1-Arf1-GDP complex is more stable (which their data do suggest). But their data do not clearly discriminate between positive or negative regulation…. If Src is a positive regulator of GBF1, then why does phospho-GBF1 form stable complexes with Arf1-GDP? A GEF that binds *more* stably to Arf1-GDP is actually *less* active. This may be counterintuitive, but this is how GEFs work: the action of BFA and the catalytic E->K mutations in Sec7 domains are both good examples of how when the GEF domain binds more strongly to Arf1-GDP, they become worse GEFs…If they were trying to demonstrate that Src is a positive regulator of GBF1, the data are not convincing.

We understand the confusion and acknowledge that perhaps the use of the word “stable” was not appropriate. We meant more “stable” in the sense of phospho-GBF1 having a higher affinity for Arf1-GDP than non-modified GBF1. This fact is supported by data from the binding assays as noted by the reviewer. One has to note that the binding assay is done in the presence of Arf-GDP, but there is no GTP so no opportunity for GEF catalysis to occur. Thus, we are only looking at the Kd (k_off_ /k_on_) of the complex.

For an enzyme, a binding affinity to the substrate is required to engage the right substrate and an increase in binding affinity does not mechanically imply a loss of catalytic activity. There are clear examples in the literature of enzymes (e.g kinases like Src) showing a decrease in Kd for their substrate and an increase of phosphorylated substrate (e.g Cbl, p130Cas, etc…).

In the case of GBF1, the protein needs to bind its Arf-GDP substrate first, then induces the change of conformation that will lead to nucleotide exchange. A classical way to describe this is a suite of reactions involving a binding reaction with a constant, k_on_ and then a catalysis reaction with the constant k_cat._ There are other constants that are important, such as the k_off_ for GBF1/Arf-GDP and k_off_ for GBF1/Arf-GTP (others, like the GBF1/Arf-GTP k_on_ can be considered negligible). Please see Author response image 1. Altogether these constants determine an apparent Kd of the GBF1/Arf complex and the rate of production of Arf-GTP.

**Author response image 1. sa2fig1:** 

The phosphorylation could in theory affect the k_on_ or the k_off_ (or both) of the GBF1/Arf-GDP complex formation. The fact that E->K mutants of GBF1 tend to form complexes with Arf1-GDP suggests that the k_off_ is low to start with. So, we can hypothesise that phosphorylation enhances the k_on_ and the complex might form more readily. (While less likely, the data could also be consistent with a change in k_off_).The k_on_ and k_cat_ are not necessarily coupled nor do they have necessarily the same magnitude. If k_on_ is larger than k_cat_, complexes can last for some time on average before the complex is being dissociated after nucleotide exchange.

If k_cat_ is larger than k_on_ (as for most enzymes in metabolic pathways acting on small molecules), the complex enzyme-substrate is very transient and barely detectable, which we believe might be what the reviewers were referring to.

The fact that GBF1 appears to form complexes with Arf-GDP, which may drive its recruitment to Golgi membranes has been proposed and documented (Quilty et al., 2014, 2018). This suggests that in the case of GBF1/Arf-GDP complex, k_on_ is larger than k_cat_ and the complex has a non-negligible half-life. But of course, this remains to be fully established.

As mentioned above, in the in vitro assay we used, the conversion to Arf-GTP is not possible, so it is mostly an increase in k_on_ / k_off_ that is measured. in vivo, we observed an increase in GBF1 recruitment at the Golgi, which is consistent with an increase in binding affinity. The decrease in Kd results in more complexes being formed and, assuming the k_cat_ is unaffected, it will also result in more Arf-GTP produced.

It is true that a loss of catalytic activity can result in an increase in binding. The difference is that in the examples cited by the reviewer, E->K mutation or BFA engagement, increased binding is not due to an increase in binding affinity but a decrease or block of catalytic activity. In other words, k_cat_ is strongly impaired or null while k_on_ is not affected.

In the modeling that was done for our study, we did not see a clear indication of stimulated catalytic activity, but rather an increase in binding affinity. Thus, our interpretation is that GBF1 phosphorylation increases the k_on_ without affecting k_cat._ Of course, it is hard to be affirmative about this without a full biochemical characterisation.

Testing this hypothesis biochemically is beyond the scope of this study. While we have attempted it, GBF1 protein is difficult to purify in significant amounts and it has not been possible to measure its GEF activity in vitro. GBF1 is a large protein prone to proteolytic degradation. It would also require various biochemical techniques for which we are not really equipped nor experts. Furthermore, and very importantly, it is likely that another protein (and maybe more) is involved in the GBF1-Arf-GDP complex formation. That protein is the receptor for Arf-GDP at the Golgi. There are also reports in the literature that other domains of GBF1 and other receptors are involved in GBF1 recruitment at the Golgi. Without identifying these additional players, the biochemical characterisation in vitro might be impossible or at least not representative.

In sum, the key message is that the scenario we propose is consistent with our data so far and there is no fundamental or conceptual incoherence in our model.

They have shown that *cellular* levels of Arf1-GTP transiently increase, but they have not shown that Golgi-localized Arf1-GTP increases, and there are other ArfGEFs in cells. The reduction of COPI localization provides further evidence that Arf1-GTP is not increasing at the Golgi.

As you may note, the presence of Arf1-GTP at the Golgi is actually not a central point of our proposal: we focus on GBF1, its phosphorylation and its role in GALNTs relocation. Nonetheless, the hypothesis that Arf-GTP is formed somewhere other than the Golgi disregards a lot of evidence and existing literature. It has been abundantly shown that Arf and GBF1 are involved in retrograde Golgi to ER traffic and that GBF1 induces the formation of Arf-GTP. Here, we show that GBF1 is recruited at the Golgi after Src activation.

We have also shown previously that Src is present at the Golgi, that its activation induces an increase of phospho-Y at the Golgi, that it induces the relocation of Golgi enzymes, the GALNTs, from the Golgi and that Arf1 is involved in the process (Bard et al., 2003; Chia et al., 2014, 2019; Gill et al., 2010). So, the data would overwhelmingly indicate that Arf1-GTP increases at the Golgi.

There are other GEF in the cell, but we show that phospho-deficient mutant forms of GBF1 are not able to induce an increase in Arf-GTP (Figure 5A). If other GEFs were involved, mutant GBF1 should not affect Arf1-GTP levels.

The fact that COPI at the Golgi is not increasing is not at all a proof that Arf-GTP is not increasing at the Golgi. While there is ample evidence that Arf-GTP can recruit COPI subunits to membranes in vitro, there is to our knowledge no proof that Arf1-GTP is sufficient to induce COPI vesicle formation in vivo. In addition, Arf-GTP has been proposed to be involved in reactions independent of COPI, see for instance (Bottanelli et al., 2017).

The key concept is that if a GEF bound strongly to GTPase-GDP, it would actually inhibit activation by GTP binding. Here is a reference from Antonny and Cherfils (and Chabre) for the specific example of a Sec7 domain mutant that binds to Arf-GDP better and therefore is a worse GEF: https://pubmed.ncbi.nlm.nih.gov/9649435/

This comment is similar to the objection raised above. This “key concept” is a misinterpretation of the solid and beautiful data presented in this paper. The ARNO mutant presented is unable to convert Arf-GDP in Arf-GTP (k_cat_ is virtually null), which means it remains bound to Arf-GDP and can dissociate with a certain k_off_, presumably quite low. This is why it is more strongly bound to Arf-GDP, as already mentioned above. There is no experiment in the paper in conditions where the conversion is blocked for both wild-type and mutant, so there is no way to say whether the affinity, the Kd for Arf-GDP has been affected.

Admittedly, in this paper, the k_cat_ is estimated at >10/sec, which is not consistent with the formation of a GEF/Arf-GDP complex lasting for several seconds. But the study is done with ARNO and the value is derived from experiments done in vitro, so the situation could be quite different for GBF1 in an in vivo context.

Here is a review from Wittenhofer: https://pubmed.ncbi.nlm.nih.gov/17540168/in which he key part about affinities is: "the affinities of the binary complexes between the G protein and either the nucleotide or its GEF are very high. In contrast, the affinities of the exchange factor for the nucleotide-bound G protein and of the nucleotide for the exchange-factor-bound G protein (the ternary complexes) are much lower.

This citation has limited to no bearing on our model: we are simply stating that the affinity of the phosphorylated GEF for Arf-GDP is higher than the non-phosphorylated one; we are not making a comparison between various forms of Arf.

This is consistent with the energetics of how all enzymes work – enzymes bind best not to their substrates or products, but to their transition states (otherwise enzymes would simply stabilize their substrates rather than performing catalysis). The nucleotide-free state of GTPases is the intermediate that is closest to the transition state of the exchange reaction. There is a general misconception that GEFs bind best to GDP-bound GTPases, but this is not true and has arisen because the so-called "GDP-locked" mutants that people have used actually have a reduced affinity for nucleotides (both GDP and GTP) and therefore tend towards being nucleotide free. Anyway, it's a common misconception and usually one that is not too important but in this case it matters for their mechanism."

Again, we are not disputing the fact that GBF1 needs to bind more efficiently to the transition state, nucleotide free form in order to function as a GEF. We are, however, strongly disputing the notion that an increased affinity for the GDP-bound form will necessarily result in a less efficient GEF.

Another reviewer wrote, "Since simply seeing more Arf1 at the Golgi does not necessarily mean that it is active, perhaps they could perform a FRAP experiment? Given that the Golgi is a large organelle, the kinetics of recovery would reflect association of ARF1 to the Golgi. If Src really increases the levels of active Arf1 at the Golgi, then it should affect the on/off-kinetics in a FRAP experiment. Or they could do Trp-fluorescence assays with recombinant Arf1 and GBF1."

The fraction of Arf-GTP and Arf amount at the Golgi increase and decrease in sync, which strongly suggest that the two processes are linked. The FRAP experiment or related ones are challenging for us at this time in part due to the public health situation, because access to the right microscope and required training is not possible.

Also, please see above the comments made above and in the Discussion about Arf-GTP. The question of whether Arf-GTP is present at the Golgi is not a key point of our model at this stage. We are not proposing that Arf-GTP specifically plays an active role in the relocation of GALNTs. In the presence of increased GBF1 expression (not phosphorylated), we observe an increase in Arf-GTP but little or no effect on GALNT relocation.

This observation led us to propose instead that GBF1 plays a key role and more specifically that the complex GBF1/Arf-GDP is critical in the formation of tubules and transport carriers. This is why the increase in affinity without an increase in catalytic activity would be important: it would allow the formation of more GBF1/Arf-GDP complexes on Golgi membranes.

We hypothesise that the catalytic exchange to GTP is a mechanism to deactivate the GBF1/Arf-GDP complex. Please note that this last part is an hypothesis at this stage. We are fully aware that we do not have sufficient data to fully support it.

Reviewer #1:In this study from the Bard lab, the authors explore the mechanism behind Src activation induced relocalization of GALNTs enzymes from Golgi to ER. They use an imidazole-inducible Src model, which generates increased O-GALNacylation. In response to Src activation, they observe GFP-GALNTs in tubules emanating from the Golgi. They also observe more Arf1-GTP in Src-activated cells. They Identify Y-phosphorylation sites in GBF1 induced by Src phosphorylation. They can confirm Src-induced phosphorylation at one of these sites (Y876) by phospho-specific antibody. They find that expression of the Y876F GBF1 mutant blocked the Src-activation induced increase in cellular Arf1-GTP levels. Expression of the Y876E/Y898E mutant dramatically reduced the amount of basal Arf1-GTP. They perform modeling studies suggesting that phosphorylation of Y876 should increase the affinity of the GBF1 Sec7d for Arf1.Their observations of Src activation are similar to the acute effects of BFA. BFA triggers formation of stable, complexes between GBF1 and Arf1-GDP. These complexes bind stably to the Golgi, but are inactive and effectively poison the GEF, reducing the amount of Arf1-GTP generated at that site. Indeed, Src activation results in slight depletion of COPI at the Golgi, and increase in GBF1 at the Golgi, similar to effects of BFA. BFA also triggers tubules emanating from the Golgi.The authors conclude with a model in which Src phosphorylation of GBF1 results in stable GBF1-Arf1 complexes that somehow generate tubules from the Golgi that traffic GALNTs enzymes back to the ER.

Thank you for the summary and the parallel made with the effects of BFA treatment. We agree with the similarities pointed out, however the analogy has its limits. We do not claim that phosphorylation induces a complex as stable as in the case of BFA treatment, but rather increases the affinity of GBF1 for Arf-GDP. The complex still dissociates after some time after conversion of Arf to GTP binding.

I do not find the overall mechanistic explanation of the observations to be convincing, and in my mind the authors have not ruled out alternative explanations. While the authors have made some interesting observations, I do not believe they have managed to correctly connect the results to a coherent and plausible mechanism.

We hope the re-written Discussion will help clarify our model.

The authors' model does not even appear to be internally consistent – on the one hand they are claiming that Src activation results in GBF1 activation of Arf1 at the Golgi. On the other hand, they are claiming that Src activation results in stable GBF1-Arf1-GDP complexes that would not be competent for activating Arf1.

As pointed in our reply above, we agree that the use of the term “stable” was misleading. We are claiming that the complexes GBF1-Arf1-GDP are competent for producing Arf1-GTP, but have enough residence time to explain the increase of GBF1 on Golgi membranes and mediate the formation of tubules.

In summary, the authors can't seem to decide on whether Src is a positive or negative regulator of GBF1, and it remains unclear whether the resulting GBF1-Arf1 complexes are actually responsible for trafficking GALNTs enzymes from the Golgi to the ER.

We find this comment unfair as we provide abundant evidence that GBF1 and Arf are involved in the relocation of GALNTs, as stated in the summary of the reviewer him/herself. The fact that the complex of the two specifically is involved derives from a careful analysis of the results and is consistent, as pointed by the reviewer, with previously published results on BFA effect, where a GBF1-Arf1-BFA complex induces the formation of tubules.

Specific issues with interpretations and experimental design:The authors find that overexpression of GBF1 increases the amount of Arf1 activated in cells in response to Src activation, but critically, they do not test the dependence of Src-responsive Arf1 activation on GBF1. Why have the authors not monitored the extent of Src-triggered Arf1 activation in GBF1-knockdown cells?

We thank the reviewer for the suggestion and have tested accordingly. We found that the loss of GBF1 by siRNA KD abolished the increase in Arf1-GTP loading over time of Src activation. Please find the data in Figure 3C-D.

The authors claim that Src activation increases the amount of Arf1 at the Golgi but the fluorescence images do not appear to support this claim. Although Arf1 is observed on the resulting tubules, this could represent its presence in non-productive membrane-bound complexes with GBF1.

This point is unclear to us. Maybe the reviewer assumes that Arf-GTP is the key driver in tubules formation? What are “non-productive membrane-bound complexes with GBF1”? We observe an increase in GBF1 at the Golgi; we propose that it is bound to Arf-GDP because of the binding assay and the previous literature.

The authors draw the unsupported conclusion that "these GBF1-Arf1-GDP complexes are directly involved in BFA-induced tubule formation." It would be straightforward for the authors to test whether GBF1 is indeed required for formation of the Src-induced tubules. Similarly, it would be even more convincing if they could also determine whether GBF1 is required for BFA-induced tubules (if this has not already been demonstrated in the literature).

First, we would like to point out that the sentence presented in brackets was not present in our text. We did not present this idea as a conclusion but as a working hypothesis in the Discussion section. The sentence read instead: “Instead, the BFA experiments and our results suggest that the GBF1-Arf1 complex induces the formation of tubules, albeit it is not clear how”.

Nonetheless, this was an interesting suggestion and we have now performed the imaging of GBF1-depleted cells and indeed we observe a much reduced formation of tubules (Figure S3G-H).

Furthermore, the authors need to test the effects of GBF1 Y/F and Y/E mutants on Src-induced tubule formation.

We have documented the effect of the GBF1 Y/F mutants on the relocation of GALNTs. Unfortunately, tubule formation is a transient and delicate event to image. The experiment proposed is thus technically difficult to perform and very time consuming. In addition, the GALNTs we have are GFP tagged, and so are the GBF1 mutants; so we would need to re-clone them, then re-derive a stable cell line.

Perhaps the observed increase in cellular Arf1-GTP levels could be due to the action of the BFA-independent GEFs, perhaps as a compensatory response to the loss of GBF1 activity induced by Src-dependent phosphorylation?

This alternative model would hardly satisfy the Occam’s razor test, would it? We think our model is much simpler and explain the data satisfactorily.

The authors make the claim "Altogether, these results indicate that phosphorylation on Tyrosines Y876 and Y898 drives an increase of affinity of GBF1 for Arf1-GDP, in turn increasing Arf1-GTP levels and promoting GALNTs relocation." Yet the phosphomimetic mutants have reduced Arf1-GTP levels! It appears more likely that phosphorylation is inducing a BFA-like effect, and not actually increasing GBF1 activation of Arf1.

Indeed, the lowered levels of Arf-GTP are initially a surprising result. However, active Src expression results after 12-18h in exactly the same effect: a reduction of Arf-GTP levels. Src activation results in a transient burst of Arf-GTP, followed by a slow decrease to levels below that of the control cells. It is not clear at present what is the mechanism at play, but we have proposed the following explanation:

“Perhaps after Src activation, either the Arf-GDP or GBF1 receptors (or both) are removed from Golgi membranes by tubules-derived carriers. This would explain why overnight Src expression, while inducing a marked GALNT relocation, also results in a reduction of Arf-GTP levels.”

As for the phosphomimetic mutants, we cannot observe the effect of their expression at short time points; but we expect they would behave similarly to a Src activation.

The authors results, both experimental and modeling, strongly suggest that phosphorylation results in stable, non-productive GBF1-Arf1 complexes that would not lead to an increase in Arf1 activation. For an exchange factor, a mutation or PTM that increases affinity for the GTPase substrate will actually result in reduced activation as the GEF needs to dissociate from the GTPase in order for GTP to stably bind.

Our results do not suggest this, as we have explained above.

A key question that is not addressed is whether Src phosphorylation of GBF1 triggers increased Golgi to ER transport or decreased ER to Golgi transport of GALNTs? Either of these situations could cause the observed response to Src activation assuming the GALNTs normally cycle between the Golgi and ER.

This is not correct. At steady state and at t=0 in the Src induction experiment, the amount of GALNTs in the ER is negligible, so blocking ER export could not have a major effect on their intracellular localisation unless there is a very rapid cycling from the Golgi to the ER. Our imaging experiment does not reveal any such transport at steady state. Finally, in addition to GBF1, all the regulators of the pathway we have identified, such as Src and ERK8, are clearly located at the Golgi (Chia et al., 2014).

Reviewer #2:In this manuscript, Chia et al. tried to provide a first glimpse of the molecular machinery driving the GALNTs Activation (GALA) pathway, which was proposed by the Bard group ten years ago. The major claim of the paper is that the regulated Src-dependent phosphorylation of GBF1 (specifically on Y876 and Y898) is the primary molecular switch that drives formation Golgi-originated membrane tubules that serve to deliver GALNT enzymes to ER. The experiments performed in the manuscript are appreciable and abundant. However, the presentation and writing are sloppy, which has made comprehension tiresome and also tricky. The discussions are shallow, and the authors have conveniently avoided rationalizing interesting/surprising findings. Overall, the hypothesis that Src activation and phosphorylation of GBF1 control specialized retrograde Golgi-ER trafficking looks exciting and valid, but this needs to be supported by more robust data, which is unfortunately not presented in this manuscript. A substantial revision would be required for this manuscript to be published.1. A major assertion throughout this manuscript is the regulated relocalization of the enzymes GALNTs from Golgi to ER, and corresponding HPL staining has been used as a readout of GANLT activity. Unfortunately, all the images of HPL staining throughout the paper are of low quality and have inconsistencies in the staining pattern and intensity. It is essential to provide high-resolution images and quantify colocalization of HPL signal with ER and Golgi markers. Colocalization of the endogenous GALNT1 with these markers will be a more accurate measure of the relocalization of the enzyme to the ER. PDI IP with VVL was a smart approach and indeed suggest GALNT activity in the ER but could also indicate minor relocalization of PDI4A to the Golgi compartment. The effect of the imidazole treatment on intracellular localization of PDI should be tested.

As the reviewer has noted, we have characterised the pathway for the last 10 years and published several studies about it. The facts that GALNTs are relocated to the ER or that the increased HPL staining is at the ER have been established before several times (Chia et al., 2014, 2019; Gill et al., 2010, 2013). They are not the focus of this study.

In F1b, colocalization of GALNT with an ER marker will substantiate this finding. This figure also shows a noticeable increase in the perinuclear (Golgi?) staining intensity of HPL, which seems to have been ignored. Does the relocalization of the enzyme to the ER have anything to do with its increased activity in the Golgi?

We have indeed observed that HPL staining also increases in the Golgi area. One likely explanation is that substrates that are O-glycosylated in the ER are trafficked to and concentrated at the cis-Golgi, leading to this staining increase. Consistent with this interpretation the HPL pattern around at the Golgi is reminiscent of ERGIC markers, that is partially ER and partially Golgi.

2. Another major inconsistency in the manuscript is related to the imidazole-inducible activity of Src8A7F. This activity is depicted as pY 4G10 blots, but these blots (Figures2E, 3A, 4A, S1B) are all different and inconclusive. Random blot parts ( 40-180 kDa, 70-180 kDa, 60-120 kDa) are shown. It is imperative to quantify Src activity. Hopefully, careful Src quantification will shed some light on the unexplainable temporal activation of Arf1 protein.

We mostly present pY 4G10 blots >40kDa to reduce the size of the blots so as to save space in the figure. We have presented the whole blots in the source data images. The increase in Src activity is reflected by the general increase in pY staining levels and is apparent in the blots presented.

The temporal activity of Arf1-GTP has nothing to do as far as we can see with Src activity.

Indeed, general phosphotyrosine levels and GBF1 phosphorylation are pretty stable over time. We have provided an alternative explanation.

3. The third major problem is related to the use of overexpressed tagged proteins utilized thought the study. This approach was more or less valid ten years ago, but now with advances of CRISPR and other gene-editing tools, it is possible to avoid (or at least control for) artifacts connected with protein overexpression and tagging. Authors show that overexpressed GALNT2-GFP and Arf1-V5 entering into the tubular compartment emerging from the Golgi (F2C), but they failed to show a similar pattern for the endogenous GALNT1 (S1L). Moreover, the activation pattern of the endogenous Arf1 (F2E) is very different from the activation dynamic of Arf1-V5 (F3A). All these inconsistencies should be adequately addressed in the text.

While we appreciate the importance of CRISPR based genome editing, this study was indeed started 10 years ago, so some approaches are indeed not absolutely ideal. But we have confirmed all the main points with endogenous proteins: the phosphorylation of GBF1 by Src for instance is exhaustively confirmed. We have also shown extensively that endogenous GALNT1 is relocated out of the Golgi to the ER upon growth factor stimulation in our previous publications (Chia et al., 2019; Gill et al., 2010).

4. In S1F, how was the experiment performed? What is "control 0h" and what are the other time points grouped as "24h Imdz wash"? This figure needs to be properly labeled.

We apologise for the lack of clarity in this figure. In S1F, Hela-IS cells were treated with 5mM imidazole (imdz) for 24 hours; See “24 hour imdz treatment” for the corresponding HPL staining. The imdz was subsequently washed out for various durations from 0.5hours to 8hours; See “imdz washout'' for the corresponding HPL staining after washout for 2hours and 8hours. The HPL staining intensities was quantified in Figure S1G where the green bars represent the HPL intensity levels of cells at various durations of washout after 24 hours of imdz treatment. The blue bars represent imdz treatment over various time points. The legend was re-written to improve on the clarity.

5. The localization of Src in S1D is different than that of Src8A7F. Does this mutation affect its localization?

We have shown previously that a fraction of Src is present at the Golgi (Bard et al., 2003). The localisation of Src and Src8A7F are cytoplasmic with some Golgi fraction. If you compare the images in S1D and S 1F, they look similar.

6. S1H has cells missing ManII. Is there any effect of Src activation on ManII?

Based on the observations from our paper in 2010 (Gill et al., 2010), we did not observe relocation of Mann II in presence of active Src. These cells are stable cell lines derived from antibiotic selection for more than 2 weeks against cells that do not express MannII-GFP. It is possible that some wildtype cells that did not express MannII-GFP managed to escape antibiotic selection.

7. In F1D is a smart approach to measure GALNT activity in the ER. An increase in glycosylation of PDI with time upon Src activation, but in later, it is shown that Aft-GTP bursts occur within 5-10mins after Src activation and in SrcEG mutant, Arf is depleted. This data is at odds with the prolonged effect of Src activation on the HPL staining intensity and GALNT relocalization. Moreover, the burst of GBF1 phosphorylation is seen at 120 mins in F4AandB. How do all this fit in into the prolonged effect on GALNT activity and increased HPL staining?

There is indeed a complex kinetic effect, we first observe a burst of GALNTs relocalisation and Arf-GTP levels. While the numbers of tubules then decrease, they do not return to normal, indicating that relocalisation continues but at a slower pace. Indeed, as we have described, the amount of GALNTs being relocated is not equivalent to the whole pool in the Golgi, but only a fraction of it. As GALNTs are enzymes, they continuously catalyse the addition of GalNAc sugar on substrates in the ER. This results in the increased HPL (staining O-GalNAc glycan) over time after GALNTs have started relocating. Regarding Arf-GTP, as we have mentioned, the levels of Arf-GTP may not be essential for relocation. We found GBF1 phosphorylation dependent on the Src activity and phosphorylation levels remain high with Src activation.

8. Overexpressed GALNT2 indeed shows tubules, but the endogenous GALNT1 tubules are absent in S1L. In this figure, βCOP staining is also very odd. The localization of βCOP to the Golgi is well established, and there are quite a few suitable antibodies that work well (https://doi.org/10.1111/j.1600-0854.2008.00724.x). Such poorly quality staining cannot be used to rule out the involvement of COPI in GALNT relocalization.

The experiment requires fixed cells and co-staining βCOP and GALNT2. However, due to a number of technical difficulties, it is difficult to observe endogenous GALNT tubules on fixed cells:

1. Tubules are transient in nature, as one can observe from the live imaging. This makes it harder to observe with endogenous levels of proteins.

2. The GALNT antibody that we have has weak immunofluorescence staining and is not as robust as with overexpressed GALNT2-GFP.

3. Chemical fixation with 4% paraformaldehyde (PFA) significantly disrupts tubule integrity and this was also reported in (Bottanelli et al., 2017).

The lack of involvement of COPI is not the main point of our study and we are not “ruling out” COPI. We show that we are just not observing any Golgi recruitment. We are not the first ones to make this disturbing observation: a carefully crafted study does not report any COPI coated vesicles involved in retrograde transport (Bottanelli et al., 2017). We have added some representative BCOP images from the automated high throughput microscopy that was used for quantification.

9. For F2, which shows GALNT1+2, Arf1 and Afr3 KDs provide data on the efficiency of the KD. Also, look at the effect of Afr1 KD on the localization of active GBF1. Is Arf1-GDP required for the membrane recruitment of GBF1?

It has been proposed and documented that it is Arf-GDP and a Golgi receptor that drives GBF1 recruitment to Golgi membranes by forming a complex at the Golgi (Quilty et al., 2014, 2018). With increased affinity of phosphorylated GBF1 for Arf1-GDP, it results in the increased recruitment of GBF1 to the Golgi. This is the likely explanation for what we observed upon Src induction.

10. Arf-V5 and endogenous Afr do not follow the same trend. Please repeat GGA IP in F3A and probe for endogenous Arf.

We are uncertain on why the reviewer thinks that Arf1-V5 and endogenous Arf1 did not follow the same trend. As observed in Figures 1E and 3A, we could observe the burst of GTP exchange within 5-10mins of Src activation and starts to fall after 20mins for both exogenous and endogenous A rf1.

11. Is GBF1 the only Arf GEF that is phosphorylated upon Src activation? Is it not likely that BIGI, ARNO and BRAG2's conserved tyrosine residues would be phosphorylated too once Src is activated and these activated GEFs would have some contribution to Golgi trafficking?

It is possible that Src could phosphorylate other ARFGEFs as it is a promiscuous kinase and there are pY sites on BIGs curated on the phosphosite database. Testing if Src phosphorylates other ARFGEFs is beyond the scope of the paper. We focused on GBF1 as the primary GEF regulating ER-Golgi traffic. Depletion of GBF1 blocks GALNTs relocation.

12. In F4E, Src has been claimed to directly phosphorylate GBF1. However, during the purification of GFP-GBF1 there could be other associated proteins that may be effectors of activated Src. This needs to be acknowledged and before claiming that Src kinase directly phosphorylated GBF1.

We disagree as we have several lines of evidence to show the directness of the phosphorylation. In Figure 4E, we have performed an in vitro kinase assay where recombinant Src was added to immunoprecipitated GBF1. The coomassie gel (Figure S4C) and pY antibody staining (Figure 4K) clearly reflects the immunoprecipitated GFP-GBF1 band on western blot and no other band. We have also verified by mass spectrometry that GBF1 is phosphorylated by active Src (Figure 4F). In cells, Src co-immunoprecipitated with GBF1 (Figure S4A), indicating direct binding between Src and GBF1.

13. Another significant discrepancy in this study is the effect of the different Src mutants on the HPL staining and Src localization/profile itself. The model in figure 7 fails to explain how the short burst of pGBF1 and AFR1-GTP triggers prolonged relocalization of GALNT. The final model is unclear and somewhat misleading. It seems to indicate that tubule formation is caused by the tight Arf1-GBF1 complex, but the data show a transient burst of Arf1-GTP (not bound to GBF1) as a major cause of tubule formation. What selects GALNTs into these tubules.

The data does not show that ARF-GTP is a major cause of tubule formation. The presence of the burst does not necessarily indicate that it is involved in tubules formation. Regarding the selection of GALNTs, we do not know the mechanism.

Reviewer #3:The manuscript by Chia et al. is one of a series of elegant manuscripts by the Bard lab on the GALA pathway. The topic is very interesting and I think it fits to the scope of the journal. The data are mostly of very high quality and the finding is novel, as it explains the role of Src in retrograde transport.

We thank reviewer 3 for the kind comments, much appreciated.

Apart from some technical and minor comments that I mention below, I am mainly concerned with one point: the authors claim that Src induces retrograde transport that is dependent on Arf1 and GBF1. However, the claim that it is COPI-independent. The evidence for this is relatively weak. Regardless of this, there is no evidence that the tubules that the tubules observed in response to Src activation are directed towards the ER. I did not see evidence that these tubules move towards the ER, or whether they passage through the ERGIC. Maybe these tubules detach from the Golgi to form a carrier that moves to the ER.I am happy to drop this point, if the editor and/or the other reviewers think that this is beyond the scope of this paper.

We agree that COPI independence is not formally established in our manuscript. We wrote that our data suggests it, we do not claim proof. This is a difficult point to prove as COPI is a key element for Golgi integrity and even required for cell viability. However, please note that this is not a central point of the manuscript.

Regarding the role of the tubules, it is correct that we did not observe fusion with ER, but this is a standard result in the field (Bottanelli et al., 2017; Sciaky et al., 1997; Sengupta et al., 2015). We show that the enzymes relocate from the Golgi to the ER, that they exit the Golgi in tubules and that we cannot detect any vesicles at the time of relocation. The tubules are specific for GALNTs and do not contain another Golgi enzyme, so they are fitting the bill of transport carrier precursors.

1. The images in Figures 1B and S1C are identical. This should be clearly indicated. The authors should state that S1C shows the very same cells as 1B, only with a co-staining for Giantin.

We have indicated in the legend of S1C that the images are from Figure 1B. Thank you for pointing this out.

2. I noticed that the size of the Golgi (and the cell) is bigger in imidazole-treated cells (with Src activation). Since the increase of fluorescence is mostly apparent in the Golgi region, I think that the HPL intensity should be normalized by the size of the cell.

We do not detect a general change in cell size upon imidazole treatment. The image of untreated cells was a cluster of cells that were more tightly packed together, hence they look smaller. We have changed the image of the untreated cells with another field of similar cell sizes for a better comparison.

3. Figure 1G: I think that the authors should image more than just 4 cells. It is also not indicated how many experiments were performed.

Each cell was acquired on different wells and on 3 different days. To further substantiate the GALNT2 tubule formation upon Src activation, our new Figure S3G demonstrates increased tubule formation in control cells (treated with non-targeting siRNA) within 30 mins of imidazole treatment.

4. Figure 2C: this result requires some form of quantification. How many tubules were observed and in how many cells? How many of the tubules were Arf1-positive?

We quantified the tubules number in Figure 1F. We did not quantify the number that are Arf positive as this is not a critical point: we are not making a statement on the Arf role but on GBF1.

5. The conservation of tyrosines (in GBF1) from yeast to mammals is meaningless. Yeast (and fungi in general) have no tyrosine kinases (there are very few exotic exceptions, but *S. cerevisiae* is definitively negative). The fact that yeast has no tyrosine kinases actually should prompt to investigate the tyrosine residues that are not conserved. I think this passage should be re-written. As it is scientifically not accurate and misleading.

Actually, while there is no tyrosine kinase per se in yeast, phosphotyrosines and corresponding kinases have been reported (Malathi et al., 1999; Stern et al., 1991). There could be as many as 27 different double specificity kinases in yeast (Zhu et al., 2000). Thus, we think our statement is scientifically accurate.

6. Figure 5: the authors state that the stimulus was "nearly abolished". This is not correct. Looking at the blot, I would. Re-word this and rather use "reduced". The double mutant is also still phosphorylated

We stand by our statement: in Figure 5B, there is no difference between 0 and 10 min imidazole, there is therefore no detectable increase of Arf-GTP.

7. Figure 5F is missing. Figure 5E just shows the quantification, but no primary data.

We have added the panel in Figure 5E and moved the quantification to 5F.

[Editors’ note: what follows is the authors’ response to the second round of review.]

The reviewers all agreed that the most critical thing missing is direct demonstration of GBF1-Arf1-GDP tubulation activity. They also suggest that you should consider alternative explanations for the data. Specifically, they suggest you load Arf1 with GDP-betaS and add phosphorylated GBF1, then test if the complex drives formation of tubules or carriers out of Golgi membranes.; they note that others have purified GBF1 and it may be possible to immunopurify a FLAG-tagged version of the protein expressed in 293T cells.Alternatively, if you can show that inducible expression of GBF1 phospho-mimetic mutants Y876E and Y898E can drive Golgi to ER localization of GALNTs in the absence of Src activation and show the cargo selectivity of such tubules, that would be sufficient to justify presentation in eLife. This alternative would require a complete reframing of the text to indicate that an unknown mechanism yet to be elucidated accomplishes this unexpected but fascinating molecular process. Such reframing would be important as the reviewers remain rather skeptical about GBF mediated membrane tubulation.

We understand the criticisms raised and we have refocused the discussion on the main points of the manuscript: how Src induces the relocation of GALNTs by direct phosphorylation of GBF1. Re the tubulation experiments, we have attempted some of these experiments, in particular purifying Golgi membranes and performing a GEF assay in these conditions (see below in Annex), but we have not been very successful and for instance unable so far to detect any GEF activity for GBF1.

On the other hand, we have been able to show that expression of GBF1 phospho-mimetic mutants Y876E and Y898E induce tubules, in striking contrast to expression of GBF1 alone (Figure 5C-D and S5C-D). We also show that depletion of GBF1 completely abolishes the formation of tubules (Figure S3G-H).

Reviewer #1:The revised version of the manuscript by Chia et al. is only a minimally altered version of the original submission. The main changes in the text are within the discussion and the authors are mainly to argue against the criticism raised by the reviewers.I find the following points confusing:1. The authors state that they don't think that it is a key point that Arf1-GTP is at the Golgi. They propose that it is the complex between GBF1 and Arf1-GDP that is relevant for tubule formation. I am not sure that the experimental evidence that is provided is convincingly showing this.

We understand the reviewer’s concerns and we have removed the model with GBF1-Arf-GDP. We have also removed most text from the discussion about the role of Arf1 and its nucleotide bound form. Instead, we focus on Src phosphorylation of GBF1. In particular, we added the results that GBF1 depletion completely abolishes tubules formation and that GBF1 phosphomimetic mutants induce their formation.

2. The authors argue that it is irrelevant to talk about COPI, because Arf1 could generate tubule itself. They cite a paper by Francesca Bottanelli. I would like to stress that the Botallei paper showed Arf1 tubules that are directed to the cell periphery and NOT retrograde transport to the ER. In addition, the Botanelli paper did not suggest that these are tubules generated with Arf1-GDP. Therefore, I find the argumentation used in the rebuttal a bit confusing.

We apologise for the confusion and the apparently poorly written previous rebuttal. We have also limited the discussion on COPI as most of our results are negative (e.g. no increase in COPI staining).

3. I think that the point that the complex of Arf1-GDP and GBF1 generates tubules should be demonstrated experimentally.

Please see above, answer to point 1.

4. The argument about the conservation of the tyrosine residue in yeast is confusing. Firstly, there is very little tyrosine phosphorylation in yeast. There are dual-specificity kinases in yeast that can perform tyrosine phosphorylation. However, there kinases are conserved in humans. So why do mammalian cells then use Src, and not the ancestral dual-specificity kinase. I find it confusing why the authors are insisting on keeping a piece of text that is so speculative and most likely wrong. Anyhow, this is just a minor point.

We acknowledge this concern and we have removed from the text the reference to GBF1 in yeast.

I think that the point that the complex of Arf1-GDP and GBF1 generates tubules should be demonstrated experimentally. Based on the new discussion and the rebuttal letter, I see that the authors themself consider this very important. I think this point deserves to be tested experimentally.

As discussed above, we have been able to show that phosphomimetic GBF1 does induce tubules, supporting the notion that phosphorylation of GBF1 by Src is what drives tubule formation. Whether it is a GBF1-Arf-GDP complex remains to be fully demonstrated and has been largely removed from the discussion.

Reviewer #2:Signaling pathways modify intracellular membrane trafficking and protein modifications on different levels. In this study, the authors investigate the mechanism behind Src activation-induced relocalization of a subset of Golgi enzymes, GALNTs, from Golgi to ER. In response to Src activation, authors observe GALNT2-GFP in Golgi-derived tubular structures. They also observed temporal activation of small GTPase Arf1 and phosphorylation of Arf1 GTP exchange factor, GBF1. The authors propose the model in which phosphorylation of GBF1 by Src results in GBF1-Arf1 complexes that generate membrane tubules for traffic GALNTs from Golgi to ER.In the revised submission, Chia et al. have fixed the many errors in the manuscript, which has indeed improved its presentation. They also provided two new experiments and significantly updated the Discussion section. At the same time, authors mostly responded to our and other critiques not by performing requested experiments/controls but by referencing their own previously published work and by modifying the text. We believe that this kind of response is not adequate.My main concerns are as follows:1. As I have stated in the first round of review, to quantify Src-dependent Golgi-ER relocalization of Golgi enzymes, it is essential to provide reproducible, high-quality images quantify colocalization of HPL signal with ER and Golgi markers. The ratio of ER to Golgi signal is the most important parameter here. The work should be reproducible, and therefore mere references to previously published work are not sufficient. The model system (HeLa cells with inducible Src) is not adequately characterized in terms of relocalization of GALNTs from Golgi to ER. Specifically, images presented in Figure 1B and, even more strikingly, in Supplementary Figure 1C, H are not supporting the notion that ER/Golgi ratio of HPL signal (i.e. relocalization of GALNTs) has been changed significantly. Without proper verification that retrograde trafficking of Golgi enzymes is increased in a Src/GBF1-dependent manner, the title of the manuscript is not supported by the data since the direction of movement of GALNT-GFP-positive tubular structures is unknown.

We apologise for not addressing fully these concerns in the previous round of revision as we had not fully appreciated the concern raised. We understand the reviewer makes reference to the fact that both ER and Golgi Tn staining increases over time after Src stimulation. We have often observed this phenomenon before. The interpretation is that after neo-synthesised proteins are Tn-glycosylated in the ER, they traffic to the Golgi, thus raising the levels of Tn in this organelle as well. (Our unpublished data indicate that over 100 cell-surface proteins are hyper-glycosylated after relocation; we have published the data for MMP14, a well-described cell surface protease.) The ER relocation results in an amplification of total Tn staining in the cell, not just in the ER but also in ERGIC and Golgi. In other words, the “ER/Golgi ratio of HPL signal” is not actually reflecting the degree of relocalisation of GALNTs. As you could see clearly in Figure 1B, we show that VVL staining in the ER increases over time; it is a clear indication that GALNTs have relocated. In addition, we complement this result with the glycosylation of ER resident proteins in Figure 1D.

2. Endogenous Golgi cargo has not been detected in Golgi-derived tubules, suggesting that tubule formation could be an artifact of protein overexpression. Authors' arguments that "tubules are transient in nature… it makes it harder to observe…chemical fixation significantly disrupts tubule integrity" are valid in general, but not at the level of eLife quality paper. Moreover, for the majority of tubule imaging in the manuscript (Figures 1G, 2C, S1K, S2A), the authors successfully used chemical fixation to demonstrate the association of overexpressed proteins with tubular structures. If necessary, consider live cell microscopy.

We thank the reviewer for the suggestion. However, we have attempted to stain endogenous GALNT with antibodies but the immunofluorescence staining with the antibodies we have is inherently weak. Together with the transient nature of tubules (i.e. the cells must be emitting tubules at the time of fixation) and the disruptive nature of chemical fixation, we are unable to clearly observe tubules with endogenous GALNT. The reviewer has suggested live cell microscopy to demonstrate the tubules. In this version of the manuscript, we present more evidence using live microscopy to demonstrate the formation of GALNT2 tubules with the GBF1-YE mutant. Altogether, we have demonstrated tubules in various settings i.e. Src-induced and GBF1-YE mutant induced, using the GALNT2-GFP stable cell line.

3. Authors clearly shown that Src phosphorylates GBF1, and they identified target phosphorylation sites on GBF1. Authors are suggesting that upon Src activation, GBF1 binds to Arf1-GDP, and this complex stimulates the formation of GALNT carrying tubules. However, I still have a hard time aligning this hypothesis with the data presented in the manuscript. During the burst of Arf-GTP, one would assume that k-on exceeds k-off resulting in Arf-GTP levels peaking and Afr-GTP exceeding GBF1-Arf-GDP. However, starting from 10 min following Src activation, the GALNT tubules emanating from the Golgi are significantly increased. This would indicate that tubule formation is not really driven by GBF1-Arf-GDP because it peaks at 20-30 min, when GBF1-Afr-GDP would be at its lowest. In S2B they show that Arf-GTP levels are lower than the controls. In S5AandB, Afr-GTP level as low as the control condition. Authors conclude that constitutively active SrcEG has the same effect on Arf-GTP levels as phosphor-mimetic GBF1 resulting in low levels of Afr-GTP. One can imagine a hypothetical scenario where the entire pool of GBF1 is engaged with Arf-GDP. But, SrcEG cells do have increased HPL staining in the ER and Golgi which, as the authors claim, is due to GALNTs transported to the ER in GBF1-Arf dependent tubules. This would indicate that the transport of GALNTs to the Golgi is independent of Arf-GTP, which is at odds with the kinetics of tubule formation and Arf-GTP levels in Figure 2.

We agree that the interpretation of the kinetics and the levels of Arf-GTP is complicated. As mentioned above, we have markedly reduced the place of the hypothesis of GBF1-Arf-GDP and removed it from the model. A full dissection of the mechanisms at play and the role of the Arf-GTP will require further study.

4. As suggested by other reviewers, to validate the model that Src-dependent phosphorylation of GBF1 is causing relocalization of Golgi enzymes, it will be essential to show that inducible expression of GBF1 phospho-mimetic mutants Y876E and Y898E would drive Golgi to ER localization of GALNTs in the absence of Src activation.

As discussed above, this has been done and while it took some optimising (relatively short expression time), we did observe significant tubule formation after GBF1 phosphomimetic expression (Figure 5C).

Reviewer #3:The authors model is now more clear, but still not convincing. They are proposing that GBF1-Arf1-GDP complexes are tubulating membranes. There is no precedent for such an activity and other plausible explanations have not been ruled out. As stated in the previous review, an alternative explanation is that their observations are similar to those observed under BFA treatment. The Hsu and Luini groups explored one possibility for why BFA induces Golgi tubulation: (https://pubmed.ncbi.nlm.nih.gov/21725317)The authors model and cartoon for the GEF reaction in Figure 7A is too simplistic, as the step they label with "kcat" actually represents more than one step: first GDP must dissociate before GTP can dissociate. This is absolutely essential as GDP and GTP occupy the same binding site. Also, the use of "kcat" generally refers to the rate-limiting step, and this is exactly the point I am making – an increase in k_on_ (which appears to be the consequence of phosphorylation) is irrelevant to the overall reaction rate constant if k_on_ is not the rate-limiting step. The step labeled "kcat" could very well be rate-limiting (and at the very least there is no reason to conclude that it is not rate-limiting, which is what the authors appear to be claiming). Therefore, my original concern still stands: their data are most consistent with phospho-GBF1 forming a stable complex with Arf1-GDP, which will reduce, rather than enhance the kinetics of exchange.I also note that in Figure 7, the authors have incorrectly used upper case 'K's, which are used for equilibrium constants, rather than lower case 'k's, which should be used for the rate constants that they are referring to. Furthermore, by convention kcat is used to refer to the overall rate constant of the reaction.The authors claim in the rebuttal letter that the "kcat" step is unlikely to be rate-limiting because this is the case for "most enzymes in metabolic pathways acting on small molecules" is both unfounded and probably irrelevant to an exchange factor.The authors claim that the Antonny, Chabre, and Cherfils paper supports their model but I strongly disagree. Yes, the mutant they used blocks exchange, and also stabilizes binding to Arf-GDP. The authors appear to be ignoring the fact that GDP must dissociate before GTP can bind. Strong binding to Arf-GDP will slow GDP dissociation, and therefore also slow the rate of exchange. The authors' strong language on these points does not make their logic any more correct.The authors are twisting themselves in knots by explaining that their in vitro binding assay does not include GTP, rather than performing an actual exchange assay in which GTP is included. Rather than trying to argue with reviewers, they could simply perform an actual nucleotide exchange experiment to see whether phosphorylated GBF1 is a better GEF or not. Based on their proposed model for how tyrosine-phosphorylation within the Sec7-domain enhances GEF activity, this should be straightforward to perform using the Sec7-domain of GBF1, rather than the full-length protein which the authors note is difficult to purify.Finally, from my perspective, I still don't understand why on the one hand, the authors are arguing that phosphorylation makes GBF1 a better GEF, yet on the other hand, the authors' model invokes a functional role for a stable GBF1-Arf1-GDP complex. Neither of these two possibilities is fully supported by the data.

We thank the reviewer for this extensive discussion. As mentioned above, we really appreciate the time and effort required to examine our arguments and we apologise if the “strong” language might have given another impression. Unfortunately, we have not been able to perform all the experiments suggested by the reviewer, in particular the purification of a Sec7 domain of GBF1 and usage in a GEF assay. As shown in Author response image 2, we have purified GBF1 and set-up a GEF assay. Unfortunately, as shown in Author response image 3, tested a GEF assay; unfortunately, we have been unable to obtain a specific GEF signal. It is not clear at present why.

**Author response image 2. sa2fig2:** Measuring purified GBF1 GEF activity using fluorescent Mant-GDP. (A) Purification of full length GST-GBF1 and phosphorylated GST-GBF1 (”GST-GBF1+SrcEG”) from Expi293T cells (See arrow for 206kDa protein). Cells were allowed to express the protein for 2 days before harvesting for purification. Gluthathione agarose beads was used to purify GST-GBF1 before 3 washes, followed by elution with 10mM Gluthaione in wash buffer. Full length GBF1 protein was eluted in eluate 1 and 2 with little or no contaminants. The protein was further washed and concentrated in an Amicon ultra centrifuge filter. (B) Schematic of Mant-GDP loading on Arf1-E17 protein as described in Kanie T. et al., 2018, Guanine Nucleotide Exchange Assay Using Fluorescent MANT-GDP. Bio Protoc. 2018 Apr 5; 8(7): e2795. (C) Loading efficiency of Mant-GDP on Arf1-E17 was ~99% efficient. The fluoescence levels of loaded Arf1 was calculated against a standard curve of free Mant-GDP fluoresence. (D) Schematic of Mant-GDP GEF activity assay as described in Kanie T. et al., 2018.

**Author response image 3. sa2fig3:** Measuring purified GBF1 GEF activity using fluorescent Mant-GDP. (A) Arf1-Mant-GDP GEF assay with purified wildtype GBF1. The rate of Mant-GDP exchange i.e. fluorescence decline of Arf1-Mant-GDP in presence of GBF1 or in control with non-hydrolyzable GTP analog GppNHp (”+GppNHp”) were similar. (B) Arf1-Mant-GDP GEF assay in presence of purified Golgi membranes. Golgi membranes were purified from HEK293FT cells using the Minute Golgi Apparatus Enrichment Kit (Invent Biotechnologies inc). There was no difference in the rate of Mant-GDP exchange between wells containing control non-hydrolyzable GTP analog GppNHp (”+GppNHp”) buffer and purified GBF1. There is miminal difference in GDP exchange between GBF1 and phospho-GBF1. The results altogether indicate the lack of functional GEF activity in the purified GBF1 protein.

However, we have been able to show that in the absence of GBF1, there is no increase in Arf-GTP, strongly suggesting that GBF1 is the GEF responsible (Figure 3C).

Regarding our model, we fully agree of course that GDP must disengage from Arf1 before GTP can bind. We are proposing that GBF1 binding to Arf-GDP must precede the nucleotide exchange reaction. We argue that this binding step can be rate limiting for the whole reaction. Supporting this idea is the over-expression of GFB1 (wt) resulting in more Arf-GTP (Figure 3A). A simple interpretation is that increasing GBF1 amounts favors complex formation. Another way to increase the rate of complex formation would be to favor a conformation of GBF1 that binds better to Arf1-GDP. This would not prevent the intramolecular rearrangements that “kick” GDP out of Arf1. We agree with the reviewer that this model does not necessarily predict a more “stable” GBF1-Arf1-GDP and we have removed this notion from the text.

[Editors' note: further revisions were suggested prior to acceptance, as described below.]

The manuscript has been improved but there are some remaining issues that need to be addressed, as outlined here.The reviewers felt that although SrcKM is supposed to be dominant negative, it is difficult to account for the effects of endogenous Src and other family members. To show that the phosphomimetic mutant of GBF1 is sufficient to induce tubule formation, the reviewers felt that the experiment should be carried out in Src-deficient cells, i.e. with pharmacological or genetic interference and not with a dominant negative approach. Reviewer 2 agreed that you would know best how to do that, as long as you can demonstrate that Src is not active under whatever treatment they use.

To improve the readability, we have re-written the parts of the paper concerning these mutants and organised them in two independent parts. We have also moved the panel with the effect of phosphomimetic GBF1 mutant on GALNT relocation into the main figure 5. It should now be clearer that GBF1 phosphomimetic effect is independent of any Src stimulation. Please see subsection “Phosphorylation at Y876 and Y898 is required for Src-induced Arf1-GTP levels and GALNT relocation” and “A Phosphomimetic mutant at Y876 and Y898 recapitulates GALNT tubule formation”.